# Operator World Models for Reinforcement Learning

**Pietro Novelli**
Istituto Italiano di Tecnologia
pietro.novelli@iit.it

**Marco Pratticò**
Istituto Italiano di Tecnologia
marco.prattico@iit.it

**Massimiliano Pontil**
Istituto Italiano di Tecnologia
AI Centre, University College London
massimiliano.pontil@iit.it

**Carlo Ciliberto**
AI Centre, University College London
c.ciliberto@ucl.ac.uk

## Abstract

Policy Mirror Descent (PMD) is a powerful and theoretically sound methodology for sequential decision-making. However, it is not directly applicable to Reinforcement Learning (RL) due to the inaccessibility of explicit action-value functions. We address this challenge by introducing a novel approach based on learning a world model of the environment using conditional mean embeddings. Leveraging tools from operator theory we derive a closed-form expression of the action-value function in terms of the world model via simple matrix operations. Combining these estimators with PMD leads to POWR, a new RL algorithm for which we prove convergence rates to the global optimum. Preliminary experiments in finite and infinite state settings support the effectiveness of our method[1].

## 1 Introduction

In recent years, Reinforcement Learning (RL) [1] has seen significant progress, with methods capable of tackling challenging applications such as robotic manipulation [2], playing Go [3] or Atari games [4] and resource management [5] to name but a few. The central challenge in RL settings is to balance the trade-off between exploration and exploitation, namely to improve upon previous policies while gathering sufficient information about the environment dynamics. Several strategies have been proposed to tackle this issue, such as Q-learning-based methods [4], policy optimization [6, 7] or actor-critics [8] to name a few. In contrast, when full information about the environment is available, sequential decision-making methods need only to focus on exploitation. Here, strategies such as policy improvement or policy iteration [9] have been thoroughly studied from both the algorithmic and theoretical standpoints. Within this context, the understanding of Policy Mirror Descent (PMD) methods has recently enjoyed a significant step forward, with results guaranteeing convergence to a global optimum with associated rates [10, 11, 12].

In their original formulation, PMD methods require explicit knowledge of the action-value functions for all policies generated during the optimization process. This is clearly inaccessible in RL applications. Recently, [12] showed how PMD convergence rates can be extended to settings in which inexact estimators of the action-value function are used (see [13] for a similar result from a regret-based perspective). The resulting convergence rates, however, depend on uniform norm bounds on the approximation error, usually guaranteed only under unrealistic and inefficient assumptions such as the availability of a (perfect) simulator to be queried on arbitrary state-action pairs. Moreover, these strategies require repeating this sampling/learning process for any policy generated by the PMD algorithm, which is computationally expensive and demands numerous interactions with the

---

[1]Code available at: github.com/CSML-IIT-UCL/powr

38th Conference on Neural Information Processing Systems (NeurIPS 2024).

environment. A natural question, therefore, is whether PMD approaches can be efficiently deployed in RL settings while enjoying the same strong theoretical guarantees.

In this work, we address these issues by proposing a novel approach to estimating the action-value function. Unlike previous methods that directly approximate the action-value function from samples, we first learn the transition operator and reward function associated with the Markov decision process (MDP). To model the transition operator, we adopt the Conditional Mean Embedding (CME) framework [14, 15]. We then leverage an operatorial characterization of the action-value function to express it in terms of these estimated quantities. This strategy draws a peculiar connection with world model methods and can be interpreted as world model learning via CMEs. The notion of world models for RL has been popularized by Ha and Schmidhuber in [16] distilling ideas from the early nineties [17, 18]. Traditional world model methods such as [16, 19] emphasize learning an implicit model of the environment in the form of a simulator. The simulator can be sampled directly in the latent representation space, which is usually of moderate dimension, resulting in a compressed and high-throughput model of the environment. This approach, however, requires extensive sampling for application to PMD and incurs into two sources of error in estimating the action-value function: model and sampling error. In contrast, CMEs can be used to estimate expectations without sampling and incur only in model error, for which learning bounds are available [20, 21]. One of our key results shows that by modeling the transition operator as a CME between suitable Sobolev spaces, we can compute estimates of the action-value function of any sufficiently smooth policy in closed form via efficient matrix operations.

Combining our estimates of the action-value function with the PMD framework we obtain a novel RL algorithm that we dub *Policy mirror descent with Operator World-models for Reinforcement learning (POWR)*. A byproduct of adopting CMEs to model the transition operator is that we can naturally extend PMD to infinite state space settings. We leverage recent advancements in characterizing the sample complexity of CME estimators to prove convergence rates for the proposed algorithm to the global maximum of the RL Problem. Our approach is similar in spirit to [22], which proposed a value iteration strategy based on CMEs. We extend these ideas to PMD strategies and refine previous results on convergence rates. Learning the transition operator with a least-squares based estimator was also recently considered in [23] and [24]. The latter proposed an optimistic strategy to prove near-optimal regret bounds in linear mixture MDP settings [25]. In contrast, in this work, we cast our problem within a linear MDP setting with possibly infinite latent dimension. We validate our approach on simple environments from the `Gym` library [26] both in finite and infinite state settings, reporting promising evidence in support of our theoretical analysis.

**Contributions**. The main contributions of this paper are: $i$) a CME-based world model framework, which enables us to generate estimators for the action-value function of a policy in closed form via matrix operations. $ii$) An (inexact) PMD algorithm combining the learned CMEs world models with mirror descent update steps to generate improved policies. $iii$) Showing that the algorithm is well-defined when learning the world model as an operator between a suitable family of Sobolev spaces. $iv$) Showing convergence rates of the proposed approach to the global maximum of the RL problem, under regularity assumptions on the MDP. $v$) Empirically testing the proposed approach in practice, comparing it with well-established baselines.

## 2 Problem Formulation and Policy Mirror Descent

We consider a Markov Decision Process (MDP) over a state space $\mathcal{X}$ and action space $\mathcal{A}$, with transition kernel $\tau$. We assume $\mathcal{X}$ and $\mathcal{A}$ to be Polish, $\tau : \Omega \to \mathcal{P}(\mathcal{X})$ to be a Borel measurable function from the joint space $\Omega = \mathcal{X} \times \mathcal{A}$ to the space $\mathcal{P}(\mathcal{X})$ of Borel probability measures on $\mathcal{X}$. We define a policy to be a Borel measurable function $\pi : \mathcal{X} \to \mathcal{P}(\mathcal{A})$. When $\mathcal{A}$ (respectively $\mathcal{X}$) is a finite set, the space $\mathcal{P}(\mathcal{A}) = \Delta(\mathcal{A}) \subseteq \mathbb{R}^{|A|}$ (respectively $\mathcal{P}(\mathcal{X}) = \Delta(\mathcal{X}) \subseteq \mathbb{R}^{|X|}$) corresponds to the probability simplex. Given a discount factor $\gamma > 0$, an initial state distribution $\nu \in \mathcal{P}(\mathcal{X})$ and a Borel measurable bounded and non-negative *reward*[2] function $r : \Omega \to \mathbb{R}$ we denote by

$$J(\pi) = \mathbb{E}_{\nu, \pi, \tau} \left[ \sum_{t=0}^{\infty} \gamma^t r(X_t, A_t) \right] \tag{1}$$

---

[2] All the discussion in this work can be extended to the case where also the rewards are random and $\tau : \Omega \to \mathcal{P}(\mathcal{X} \times \mathbb{R})$ takes values in the space of joint distributions over states and rewards $(X_{t+1}, R_t)$

the (discounted) expected return of the policy $\pi$ applied to the MDP, yielding the Markov process $(X_t, A_t)_{t\in\mathbb{N}}$, where $X_0$ is distributed according to $\nu$ and for each $t \in \mathbb{N}$ the action $A_t$ is distributed according to $\pi(\cdot|X_t)$ and $X_{t+1}$ according to $\tau(\cdot|X_t, A_t)$.

In sequential decision settings, the goal is to find the optimal policy $\pi_*$ maximizing (1) over the space of all measurable policies. In reinforcement learning, one typically assumes that knowledge of the transition $\tau$, the reward $r$, and (possibly) the starting distribution $\nu$ is not available. It is only possible to gather information about these quantities by interacting with the MDP to sample state-action pairs $(x_t, a_t)$ and corresponding rewards $r(x_t, a_t)$ and transitions $x_{t+1}$.

**Policy Mirror Descent (PMD)**. In so-called tabular settings – in which both $\mathcal{X}$ and $\mathcal{A}$ are finite sets – the policy optimization problem amounts to maximizing (1) over the space $\Pi = \Delta(\mathcal{A}) \otimes \mathbb{R}^{|\mathcal{X}|}$ of column substochastic matrices, namely matrices $M \in \mathbb{R}^{|\mathcal{A}|\times|\mathcal{X}|}$ with non-negative entries and whose columns sum up to one, namely $M^*\mathbf{1}_{\mathcal{A}} = \mathbf{1}_{\mathcal{X}}$, with $\mathbf{1}$ denoting the vector with all entries equal to one on the appropriate space. Borrowing from the convex optimization literature – where mirror descent algorithms offer a powerful approach to minimize a convex functional over a convex constraint set [27, 28] – recent work proposed to adopt mirror descent also for policy optimization, a strategy known as *policy mirror descent (PMD)* [10]. Even though the objective in (1) is not convex (or concave, since we are maximizing it), it turns out that mirror ascent can nevertheless enjoy global convergence to the maximum, with sublinear [11] or even linear rates [12], at the cost of dimension-dependent constants.

Starting from an initial policy $\pi_0$, PMD generates a sequence $(\pi_t)_{t\in\mathbb{N}}$ according to the update step

$$\pi_{t+1}(\cdot\,|\,x) = \underset{p\in\Delta(\mathcal{A})}{\operatorname{argmin}} \quad -\eta\,\langle q_{\pi_t}(\cdot, x), p\rangle + D(p, \pi_t(\cdot|x)), \tag{2}$$

for any $x \in \mathcal{X}$, with $\eta > 0$ a step size, $D$ a suitable Bregman divergence [28] and $q_\pi : \Omega \to \mathbb{R}$ the so-called *action-value* function of a policy $\pi$, see also [12, Sec. 4]. The action-value function

$$q_\pi(x, a) = \mathbb{E}\left[\sum_{t=0}^{\infty} \gamma^t r(X_t, A_t)\,\middle|\,X_0 = x, A_0 = a\right] \tag{3}$$

is the discounted return obtained by taking action $a \in \mathcal{A}$ in state $x \in \mathcal{X}$ and then following the policy $\pi$. The solution to (2) crucially depends on the choice of $D$. For example, in [10] the authors observed that if $D$ is the Kullback-Leibler divergence, PMD corresponds to the Natural Policy Gradient originally proposed in [29] while [12] showed that if $D$ is the squared euclidean distance, PMD recovers the Projected Policy Gradient method from [11].

**PMD in Reinforcement Learning**. A clear limitation to adopting PMD in RL settings is that (3) needs exact knowledge of the action-value functions $q_{\pi_t}$ associated to each iterate $\pi_t$ of the algorithm. This requires evaluating the expectation in (3), which is not possible in RL where we do not know the reward $r$ and MDP transition distribution $\tau$ in advance. While sampling strategies can be adopted to estimate $q_{\pi_t}$, a key question is how the approximation error affects PMD.

The work in [12] provides an answer to this question, extending the analysis of PMD to the case where estimates $\hat{q}_{\pi_t}$ are used in place of the true action-value function in (2). We recall here an informal version of the result for the case of sublinear convergence rates for PMD. We postpone a more rigorous statement of the theorem and its assumptions to Sec. 5, where we extend it to infinite state spaces $\mathcal{X}$.

**Theorem 1** (Inexact PMD (Sec. 5 in [12]) – Informal). *In the tabular setting, let $(\pi_t)_{t\in\mathbb{N}}$ be a sequence of policies obtained by applying the PMD update in (2) with functions $\hat{q}_{\pi_t} : \Omega \to \mathbb{R}$ in place of $q_{\pi_t}$ and $D$ a suitable Bregman divergence. For any $T \in \mathbb{N}$ and $\varepsilon > 0$, if $\|\hat{q}_{\pi_t} - q_{\pi_t}\|_\infty \leq \varepsilon$ for all $t = 1, \ldots, T$, then*

$$\max_\pi\ J(\pi) - J(\pi_T) \leq O(\varepsilon + 1/T). \tag{4}$$

Thm. 1 implies that inexact PMD retains the convergence rates of its exact counterpart, provided that the approximation error for each action-value function is of order $1/T$ in uniform norm $\|\cdot\|_\infty$. While this result supports estimating the action-value function in RL, implementing this strategy in practice poses two main challenges, even in tabular settings. First, approximating the expectation in (3) in $\|\cdot\|_\infty$ norm via sampling requires "starting" the MDP from each state $x \in \mathcal{X}$, multiple times. This

is often not possible in RL, where we do not have control over the starting distribution $\nu$. Second, repeating this sampling process to learn a $\hat{q}_{\pi_t}$ for each policy $\pi_t$ can become extremely expensive in terms of both the number of computations and interactions with the environment.

In this work, we propose a new strategy to tackle the problems above. Instead of re-sampling the MDP to estimate $\hat{q}_{\pi_t}$ at each iteration $t$, we learn estimators $\hat{r}$ and $\hat{\tau}$ for the reward and transition distribution, respectively. For any policy $\pi$, we then leverage the relation between these quantities in (3) to generate an estimator $\hat{q}$ for $q_\pi$. This approach tackles the above challenges since 1) it enables us to control the approximation error on any action-value function in terms of the approximation error of $\hat{r}$ and $\hat{\tau}$; 2) it does not require sampling the MDP to learn a new $\hat{q}_{\pi_t}$ for each $\pi_t$ generated by PMD.

## 3 Operator World Models

In this section, we present an operator-based formulation of the problem introduced in Sec. 2 (see also [11]). This will be instrumental in extending the PMD theory to arbitrary state spaces $\mathcal{X}$, to quantify the approximation error of the action-value function in terms of the approximation error of the reward and transition distribution, and to motivate conditional mean embeddings as the tool to learn these latter quantities.

**Conditional Expectation Operators**. We start by defining the *transfer operator* $\mathsf{T}$ associated with the MDP transition distribution $\tau$. Let $B_b(\mathcal{X})$ denote the space of bounded Borel measurable functions on a space $\mathcal{X}$. Formally, $\mathsf{T} : B_b(\mathcal{X}) \to B_b(\Omega)$ is the linear operator such that, for any $f \in B_b(\mathcal{X})$

$$(\mathsf{T}f)(x,a) = \int_\mathcal{X} f(x') \, \tau(dx'|x,a) = \mathbb{E}\left[f(X') \mid x,a\right] \qquad \text{for all } (x,a) \in \Omega, \qquad (5)$$

where $X'$ is sampled according to $\tau(\cdot|x,a)$. Note that $\mathsf{T}$ is the Markov operator [30, Ch. 19] encoding the dynamics of the MDP and its conjugate $\mathsf{T}^* : \mathcal{M}(\Omega) \to \mathcal{M}(\mathcal{X})$ is the operator mapping signed Borel measures $\mu \in \mathcal{M}(\Omega)$ to their transition via $\tau$ as $(\mathsf{T}^*\mu)(\mathcal{B}) = \int_{\mathcal{B} \times \Omega} \tau(dx'|x,a)\mu(dx,da)$ for any measurable $\mathcal{B} \subseteq \mathcal{X}$. For any policy $\pi$ we define the operator $\mathsf{P}_\pi : B_b(\Omega) \to B_b(\mathcal{X})$ such that for all $g \in B_b(\Omega)$

$$(\mathsf{P}_\pi g)(x) = \int_\mathcal{A} g(x,a) \, \pi(da|x) = \mathbb{E}\left[g(X,A) \mid X = x\right] \quad \text{for all } x \in \mathcal{X}, \qquad (6)$$

where the expectation is taken over the action $A$ sampled according to $\pi(\cdot|x)$. Also $\mathsf{P}_\pi$ is a Markov operator and its conjugate $\mathsf{P}_\pi^* : \mathcal{M}(\mathcal{X}) \to \mathcal{M}(\Omega)$ is the operator mapping any $\nu \in \mathcal{M}(\mathcal{X})$ to its joint measure with $\pi$, namely $(\mathsf{P}_\pi^*\nu)(\mathcal{C}) = \int_\mathcal{C} \pi(da|x)\nu(dx)$ for any measurable $\mathcal{C} \subseteq \Omega$.

**Operator Formulation of RL**. With these two operators in place, we can characterize the expected reward after a single interaction between a policy $\pi$ and the MDP as $(\mathsf{T}\mathsf{P}_\pi r)(x,a) = \mathbb{E}[r(X',A')|X_0 = x, A_0 = a]$. This observation can be applied recursively, yielding the operatorial characterization of the action-value function from (3)

$$q_\pi(x,a) = \sum_{t=0}^\infty \gamma^t \mathbb{E}[r(X_t,A_t)|X_0 = x, A_0 = a] = \sum_{t=0}^\infty (\gamma\mathsf{T}\mathsf{P}_\pi)^t r = (\mathsf{Id} - \gamma\mathsf{T}\mathsf{P}_\pi)^{-1}r, \qquad (7)$$

where the last equality follows from $\mathsf{T}$ and $\mathsf{P}_\pi$ being Markov operators [30, Ch. 19] ($\|\mathsf{T}\| = \|\mathsf{P}_\pi\| = 1$), making the Neumann series convergent. Analogously, we can reformulate the RL objective introduced in (1) as the pairing

$$J(\pi) = \left\langle \mathsf{P}_\pi(\mathsf{Id} - \gamma\mathsf{T}\mathsf{P}_\pi)^{-1}r, \nu \right\rangle = \left\langle \mathsf{P}_\pi q_\pi, \nu \right\rangle, \qquad (8)$$

for $\nu \in \mathcal{P}(\mathcal{X})$ a starting distribution. In both (7) and (8) the operatorial formulation encodes the cumulative reward collected through the (possibly infinitely many) interactions of the policy with the MDP in closed form, as the inversion $(\mathsf{Id} - \gamma\mathsf{T}\mathsf{P}_\pi)^{-1}r$. This characterization motivates us to learn $\mathsf{T}$ and $r$ from data and then express any action-value function as the interaction of these two terms with the policy $\pi$ as in (7), rather than learning each $q_\pi$ independently for any $\pi$.

**Learning the World Model via Conditional Mean Embeddings**. Conditional Mean Embeddings (CME) offer an effective tool to model and learn conditional expectation operators from data [15]. They cast the problem of learning $\mathsf{T}$ by studying the restriction of its action on a suitable family

of functions. Let $\varphi : \mathcal{X} \to \mathcal{F}$ and $\psi : \Omega \to \mathcal{G}$ two feature maps with values into the Hilbert spaces $\mathcal{F}$ and $\mathcal{G}$. With some abuse of notation (which is justified by them being Hilbert spaces), we interpret $\mathcal{F}$ and $\mathcal{G}$ as subspaces of functions in $B_b(\mathcal{X})$ and $B_b(\Omega)$ of the form $f(x) = \langle f, \varphi(x) \rangle$ and $g(x, a) = \langle g, \psi(x, a) \rangle$ for any $f \in \mathcal{F}$ and $g \in \mathcal{G}$ and any $(x, a) \in \Omega$. We say that the *linear MDP* assumption holds with respect to $(\varphi, \psi)$ if

**Assumption 1** (Linear MDP – Well-specified CME). *The restriction of* $\mathsf{T}$ *to* $\mathcal{F}$ *is a Hilbert-Schmidt operator* $\mathsf{T}|_{\mathcal{F}} \in \mathsf{HS}(\mathcal{F}, \mathcal{G})$.

In CME settings, the assumption above is known as requiring the CME of $\tau$ to be *well-specified*. The following result, proved in Appendix A.2, clarifies this aspect and establishes the relation of Asm. 1 with the standard definition of linear MDP.

**Proposition 2** (Well-specified CME). *Under Asm. 1,* $(\mathsf{T}|_{\mathcal{F}})^* = (\mathsf{T}^*)|_{\mathcal{G}}$ *and, for any* $(x, a) \in \Omega$

$$(\mathsf{T}|_{\mathcal{F}})^* \psi(x, a) = \int_{\mathcal{X}} \varphi(x') \, \tau(x'|x, a) = \mathbb{E}[\varphi(X')|X = x, A = a]. \tag{9}$$

Proposition 2 shows that (9) is equivalent to the standard linear MDP assumption [31, Ch. 8] when $\mathcal{X}$ is a finite set (taking $\varphi$ the one-hot encoding) while being weaker in infinite settings. From the CME perspective, the proposition characterizes the action of $(\mathsf{T}|_{\mathcal{F}})^*$ as sending evaluation vectors in $\mathcal{G}$ to the conditional expectation of evaluation vectors in $\mathcal{F}$ with respect to $\tau$, the definition of conditional mean embedding of $\tau$ [32, 15]. This characterization also suggests a learning strategy: (9) characterizes the action of $\mathsf{T}$ as evaluating the conditional expectation of a vector $\varphi(X')$ given $(x, a)$. Given a set of points $(x_i, a_i)_{i=1}^n$ and corresponding $x'$ sampled from $\tau(\cdot|x_i, a_i)$, this can be learned by minimizing the squared loss, yielding the estimator (see [15, Sec 4.2])

$$\mathsf{T}_n = \operatorname*{argmin}_{\mathsf{T} \in \mathsf{HS}(\mathcal{F}, \mathcal{G})} \frac{1}{n} \sum_{i=1}^n \|\varphi(x_i') - \mathsf{T}^* \psi(x_i, a_i)\|_{\mathcal{F}}^2 + \lambda \|\mathsf{T}\|_{\mathsf{HS}}^2 = S_n^* K_\lambda^{-1} Z_n. \tag{10}$$

When $\mathcal{F}$ and $\mathcal{G}$ are finite dimensional, $S_n$ and $Z_n$ are matrices with $n$ rows, each corresponding respectively to the vectors $\psi(x_i, a_i)$ and $\varphi(x_i')$ for $i = 1, \dots, n$. In the infinite setting, they generalize to operators $S_n : \mathcal{G} \to \mathbb{R}^n$ and $Z_n : \mathcal{F} \to \mathbb{R}^n$. The matrix $K_\lambda = S_n S_n^* + n\lambda \mathsf{Id}_n \in \mathbb{R}^{n \times n}$ is the regularized Gram (or kernel) matrix with $(i, j)$-th entry corresponding to

$$\left(K_\lambda\right)_{ij} = \langle \psi(x_i, a_i), \psi(x_j, a_j) \rangle + n\lambda \delta_{ij}. \tag{11}$$

We conclude our discussion on learning world models via CMEs by noting that in most RL settings, the reward function is unknown, too. Analogously to what we have described for $\mathsf{T}_n$ and following the standard practice in supervised settings, we can learn an estimator for $r$ solving a problem akin to (10). This yields a function of the form $r_n = S_n^* b = \sum_{i=1}^n b_i \, \psi(x_i, a_i)$ as the linear combination of the embedded training points with the entries of the vector $b = K_\lambda^{-1} y$ where $y \in \mathbb{R}^n$ is the vector with entries $y_i = r(x_i, a_i)$.

**Estimating the Action-value Function** $q_\pi$. We now propose our strategy to generate an estimator for the action-value function $q_\pi$ of a given policy $\pi$ in terms of an estimator for the reward $r$ and a world model for $\mathsf{T}$ learned in terms of the restriction to $\mathcal{G}$ and $\mathcal{F}$. To this end, we need to introduce the notion of compatibility between a policy $\pi$ and the pair $(\mathcal{G}, \mathcal{F})$.

**Definition 1** (($\mathcal{G}, \mathcal{F}$)-compatibility). *A policy* $\pi : \mathcal{X} \to \mathcal{P}(\mathcal{A})$ *is compatible with two subspaces* $\mathcal{F} \subseteq B_b(\mathcal{X})$ *and* $\mathcal{G} \subseteq B_b(\Omega)$ *if the restriction* $\mathsf{P}_\pi$ *to* $\mathcal{G}$ *is a bounded linear operator with range* $\subseteq \mathcal{F}$, *that is* $(\mathsf{P}_\pi)|_{\mathcal{G}} : \mathcal{G} \to \mathcal{F}$.

Definition 1 is analogous to the linear MDP Asm. 1 in that it requires the restriction of $\mathsf{P}_\pi$ to $\mathcal{G}$ to take values in the associated space $\mathcal{F}$. However, it is slightly weaker since it requires this restriction to be bounded (and linear) rather than being an HS operator. We will discuss in Sec. 4 how this difference will allow us to show that a wide range of policies (in particular those generated by our POWR method) is $(\mathcal{G}, \mathcal{F})$-compatible for our choice of function spaces. Definition 1 is the key condition that enables us to generate an estimator for $q_\pi$, as characterized by the following result.

---

**Algorithm 1** POWR: POLICY MIRROR DESCENT WITH OPERATOR WORLD-MODELS FOR RL

---

**Input:** Dataset $(x_i, a_i, x_i', r_i)_{i=1}^n$, discount factor $\gamma \in (0,1)$, step size $\eta > 0$, kernel function $k(x, x') = \langle \phi(x), \phi(x') \rangle$ with $\phi : \mathcal{X} \to \mathcal{H}$ as in Proposition 4, initial weights $C_0 = 0 \in \mathbb{R}^{n \times |\mathcal{A}|}$.

```
/* World Model Learning */
```
**let** $E \in \mathbb{R}^{n \times |\mathcal{A}|}$ with rows $E_i = \text{ONEHOT}_{|\mathcal{A}|}(a_i)$.
**let** $K_\lambda \in \mathbb{R}^{n \times n}$ such that $K_{ij} = k(x_i, x_j)\delta_{a_i = a_j} + n\lambda\delta_{ij}$          ▷ Eq. (11)
**let** $H \in \mathbb{R}^{n \times n}$ such that $H_{ij} = k(x_i', x_j)$
**compute** $K_\lambda^{-1}$ and $b = K_\lambda^{-1} y$ with $y = (r_1, \ldots, r_n) \in \mathbb{R}^n$          ▷ Eq. (10)

```
/* Policy Mirror Descent */
```
**for** $t = 0, 1, \ldots, T - 1$ **do**:
     $\pi_{t+1} = \text{SOFTMAX}(\eta H C_t) \in \mathbb{R}^{n \times |\mathcal{A}|}$          ▷ PMD Step (15)
     $M_{\pi_{t+1}} = H \odot (\pi_{t+1} E^\top) \in \mathbb{R}^{n \times n}$          ▷ Proposition 3, Eq. (13)
     $C_{t+1} = C_t + \text{diag}(c)E$ with $c = (\text{Id} - \gamma K_\lambda^{-1} M_{\pi_{t+1}})^{-1} b$      ▷ Proposition 3, Eq. (12)
**end for**

**return** $\pi_T : \mathcal{X} \to \Delta(\mathcal{A})$ such that $\pi_T(x) = \text{SOFTMAX}(\eta\, H_x C_T)$ with $H_x = (k(x, x_i))_{i=1}^n \in \mathbb{R}^n$.

---

**Proposition 3.** *Let* $\mathsf{T}_n = S_n^* B Z_n \in \mathsf{HS}(\mathcal{F}, \mathcal{G})$ *and* $r_n = S_n^* b \in \mathcal{G}$ *for respectively a* $B \in \mathbb{R}^{n \times n}$ *and* $b \in \mathbb{R}^n$. *Let* $\pi$ *be* $(\mathcal{G}, \mathcal{F})$-*compatible. Then,*

$$\hat{q}_\pi = (\text{Id} - \gamma \mathsf{T}_n \mathsf{P}_\pi)^{-1} r_n = S_n^*(\text{Id} - \gamma B M_\pi)^{-1} b \qquad (12)$$

*where* $M_\pi = Z_n \mathsf{P}_\pi S_n^* \in \mathbb{R}^{n \times n}$ *is the matrix with entries*

$$\left( M_\pi \right)_{ij} = \langle \varphi(x_i'), \mathsf{P}_\pi \psi(x_j, a_j) \rangle = \int_{\mathcal{A}} \langle \psi(x_i', a), \psi(x_j, a_j) \rangle \ \pi(da|x_i'). \qquad (13)$$

Proposition 3 leverages a kernel trick argument to express the estimator for the action-value function of $\pi$ as the linear combination $\hat{q}_\pi = \sum_{i=1}^n c_i \psi(x_i, a_i)$ of the (embedded) training points $\psi(x_i, a_i)$ and the entries $c_i$ of the vector $c = (\text{Id} - \gamma B M_\pi)^{-1} b \in \mathbb{R}^n$. We prove the result in Appendix A.4. We note that in (12) both $B$ and $M_\pi$ are $n \times n$ matrices and therefore the characterization of $\hat{q}_\pi$ amounts to solving a $n \times n$ linear system. For settings where $n$ is large, one can adopt random projection methods such as Nyström approximation to learn $\mathsf{T}_n$ and $r_n$ [33]. These strategies have been recently shown to significantly reduce the computational load of learning while retaining the same empirical and theoretical performance as their non-approximated counterparts [34, 35].

We conclude this section noting how (13) implies that we only need to be able to evaluate $\pi$, but we do not need explicit knowledge of $\mathsf{P}_\pi$ as operator. As we shall see, this property will be instrumental to prove generalization bounds for our proposed PMD algorithm in Sec. 4.

## 4 Proposed Algorithm: POWR

We are ready to describe our algorithm for world model-based PMD. In the following, we restrict to the case where $|\mathcal{A}| < \infty$ is a finite set. As introduced in Sec. 2, policy mirror descent methods are mainly characterized by the choice of Bregman divergence $D$ used for the mirror descent update and, in the case of inexact methods, the estimator $\hat{q}_{\pi_t}$ of the action-value function $q_{\pi_t}$ for the intermediate policies generated by the algorithm.

In POWR, we combine the CME world model presented in Sec. 3 with mirror descent steps using the Kullback-Leibler divergence $D_{\text{KL}}(p; p') = \sum_{a \in \mathcal{A}} p_a \log(p_a/p_a')$ in the update of (2). It was shown in [10] that in this case PMD corresponds to Natural Policy Gradient [29]. As showed in [36, Example 9.10], the solution to (2) can be written in closed form for any $x \in \mathcal{X}$ as

$$\pi_{t+1}(\cdot|x) = \frac{\pi_t(\cdot|x) e^{\eta \hat{q}_{\pi_t}(x, \cdot)}}{\sum_{a \in \mathcal{A}} \pi_t(a|x) e^{\eta \hat{q}_{\pi_t}(x, a)}}, \qquad (14)$$

where we used the estimated action-value function $\hat{q}_\pi$ from Proposition 3. Additionally, the formula above can be applied recursively, expressing $\pi_{t+1}$ as the softmax operator applied to the discounted

sum of the action-value functions up to the current iteration

$$\pi_{t+1}(\cdot|x) = \text{SOFTMAX}\left(\log(\pi_0(\cdot|x)) + \eta \sum_{s=0}^{t} \hat{q}_{\pi_s}(x, \cdot)\right). \tag{15}$$

**Choice of the Feature Maps**. A key question to address before adopting the action-value estimators introduced in Sec. 3 is choosing the two spaces $\mathcal{F}$ and $\mathcal{G}$ to perform world model learning. Specifically, to apply Proposition 3 and obtain proper estimators $\hat{q}_{\pi_t}$, we need to guarantee that all policies generated by the PMD update (14) are $(\mathcal{G}, \mathcal{F})$-compatible (Definition 1). The following result describes a suitable family of such spaces.

**Proposition 4** (Separable Spaces). *Let $\phi : \mathcal{X} \to \mathcal{H}$ be a feature map into a Hilbert space $\mathcal{H}$. Let $\mathcal{F} = \mathcal{H} \otimes \mathcal{H}$ and $\mathcal{G} = \mathbb{R}^{|\mathcal{A}|} \otimes \mathcal{H}$ with feature maps respectively $\varphi(x) = \phi(x) \otimes \phi(x)$ and $\psi(x, a) = \phi(x) \otimes e_a$, with $e_a \in \mathbb{R}^{|\mathcal{A}|}$ the one-hot encoding of action $a \in \mathcal{A}$. Let $\pi : \mathcal{X} \to \Delta(\mathcal{A})$ be a policy such that $\pi(a|\cdot) = \langle p_a, \phi(\cdot)\rangle$ with $p_a \in \mathcal{H}$ for any $a \in \mathcal{A}$. Then, $\pi$ is $(\mathcal{G}, \mathcal{F})$-compatible.*

Proposition 4 (proof in Appendix A.5) states that for the specific choice of function spaces $\mathcal{F} = \mathcal{H} \otimes \mathcal{H}$ and $\mathcal{G} = \mathbb{R}^{|\mathcal{A}|} \otimes \mathcal{H}$, *we can guarantee $(\mathcal{G}, \mathcal{F})$-compatibility, provided that $\mathcal{H}$ is rich enough to "contain" all $\pi(a|\cdot)$ for $a \in \mathcal{A}$.* We postpone the discussion on identifying a suitable spaces $\mathcal{H}$ for PMD to Sec. 5, since $(\mathcal{G}, \mathcal{F})$-compatibility is not needed to mechanically apply Proposition 3 and obtain an estimator $\hat{q}_\pi$. This is because (12) exploits a kernel-trick to bypass the need to know $P_\pi$ explicitly and rather requires only to be able to evaluate $\pi$ on the training data. The latter is possible for $\pi_{t+1}$, thanks to the explicit form of the PMD update in (14). We can therefore present our algorithm.

**POWR**. Alg. 1 describes *Policy mirror descent with Operator World-models for Reinforcement learning (POWR)*. Following the intuition of Proposition 4, the algorithm assumes to work with separable spaces. During an initial phase, we learn the world model $T_n = S_n^* K_\lambda^{-1} Z_n$ and the reward $r_n = S_n^* b$ fitting the conditional mean embedding described in (10) on a dataset $(x_i, a_i, x_i', r_i)_{i=1}^n$ (e.g. obtained via experience replay [37]). Once the world model has been learned, we optimize the policy and perform PMD iterations via (15). In this second phase, we first evaluate the past (cumulative) action-value estimators $\sum_{s=0}^{t} \hat{q}_{\pi_s}$ to obtain the policy $\pi_{t+1}(\cdot|x_i)$ by (inexact) PMD via the softmax operator in (15). We use the newly obtained policy to compute the matrix $M_{\pi_{t+1}}$ defined in (13), which is a key component to obtain the estimator $\hat{q}_{\pi_{t+1}}$ for $q_{\pi_{t+1}}$. We note that in the case of the separable spaces of Proposition 4, this matrix reduces to the $n \times n$ matrix with entries $k(x_i', x_j)\pi_{t+1}(a_j|x_i')$, where $k(x_i', x_j) = \langle \phi(x_i'), \phi(x_j)\rangle$ is the kernel matrix between initial and evolved states. Finally, we obtain $c = (\text{Id} - \gamma K_\lambda^{-1} M)^{-1} b$ and model $\hat{q}_{\pi_{t+1}} = S_n^* c$ according to Proposition 3.

Clearly, the world model learning and PMD phases can be alternated in POWR, essentially finding a trade-off between exploration and exploitation. This could possibly lead to a refinement of the world model as more observations are integrated into the estimator. While in this work we do not investigate the effects of such alternating strategy, Thm. 7 offers relevant insights in this sense. The result characterizes the behavior of the PMD algorithm when combined with a varying (possibly increasing) accuracy in the estimation of the action-valued function (see Sec. 5 for more details).

## 5 Theoretical Analysis

We now show that POWR converges to the global maximizer of the RL problem in (1). To this end, we first identify a family of function spaces guaranteed to be compatible with the policies generated by Alg. 1. Then, we provide an extension of the result in [12] for inexact PMD to infinite state spaces $\mathcal{X}$, showing the impact of the action-value approximation error on the convergence rates. For the estimator introduced in Proposition 3, we relate this error to the approximation errors of $T_n$ and $r_n$ leveraging the Simulation Lemma A.6. Finally, we use recent advancements in the characterization of CMEs' fast learning rates to bound the sample complexity of these latter quantities, yielding error bounds for POWR.

**POWR is Well-defined**. To properly apply Proposition 3 to estimate the action-value function of any PMD iterate $\pi_t$, we need to guarantee that every iterate belongs to the space $\mathcal{H}$ according to Proposition 4. The following result provides such a family of spaces.

**Theorem 5.** *Let $\mathcal{X} \subset \mathbb{R}^d$ be a compact set and let $\mathcal{H} = W^{2,s}(\mathcal{X})$ be the Sobolev space of smoothness $s > 0$ (see e.g. [38]). Let $\pi_t(a|\cdot)$ and $\hat{q}_{\pi_t}(\cdot, a)$ belong to $\mathcal{H}$ for any $a \in \mathcal{A}$ and $\pi_t(a|x) > 0$ for any $x \in \mathcal{X}$. Then the policy $\pi_{t+1}$ solution to the PMD update in (14) belongs to $\mathcal{H}$.*

According to Thm. 5, Sobolev spaces offer a viable choice for compatibility with PMD-generated policies. This observation is further supported by the fact that Sobolev spaces of smoothness $s > d/2$ are so-called *reproducing kernel Hilbert spaces (rkhs)* (see e.g. [39, Ch. 10]). We recall that rkhs are always naturally equipped with a $\phi : \mathcal{X} \to \mathcal{H}$ such that the inner product $\langle \phi(x), \phi(x') \rangle = k(x, x')$ defines a so-called reproducing kernel, namely a positive definite function that is (usually) efficient to evaluate, even if $\phi(x)$ is high or infinite dimensional. For example, $\mathcal{H} = W^{2,s}(\mathcal{X})$ with $s = \lceil \frac{d}{2} \rceil$ has associated kernel $k(x, x') = e^{-\|x-x'\|/\sigma}$ with bandwidth $\sigma > 0$ [39]. By applying Thm. 5 to the iterates generated by Alg. 1 we have the following result.

**Corollary 6.** *With the hypothesis of Proposition 4 let $\mathcal{H} = W^{2,s}(\mathcal{X})$ with $s > d/2$. Let $\mathsf{T}_n \in \mathsf{HS}(\mathcal{F}, \mathcal{G})$ and $r_n \in \mathcal{G}$ characterized as in Proposition 3. Let $\pi_0(a|\cdot) \propto e^{\eta q_0(\cdot, a)}$ for $q_0$ such that $q_0(\cdot, a) \in \mathcal{H}$ any $a \in \mathcal{A}$. Then, for any $t \in \mathbb{N}$ the PMD iterates $\pi_t$ generated by Alg. 1 are such that $\pi_t(a|\cdot) \in \mathcal{H}$ and hence are $(\mathcal{G}, \mathcal{F})$-compatible.*

The above corollary guarantees us that if we are able to learn our estimates for the action-value function in a suitably regular Sobolev space $\mathcal{H}$, then POWR is well-defined. This is a necessary condition to then being able to study its theoretical behavior in our main result. We report the proofs of Thm. 5 and Cor. 6 in Appendix C.1.

**Inexact PMD Converges**. We now present a more rigorous version of the characterization of the convergence rates of the inexact PMD algorithm discussed informally in Thm. 1.

**Theorem 7** (Convergenge of Inexact PMD). *Let $(\pi_t)_{t \in \mathbb{N}}$ be a sequence of policies generated by Alg. 1 that are all $(\mathcal{G}, \mathcal{F})$-compatible. If the action-value functions $\hat{q}_{\pi_t}$ are estimated with an error $\|q_{\pi_t} - \hat{q}_{\pi_t}\|_\infty \leq \varepsilon_t$, the iterates of Alg. 1 converge to the optimal policy as*

$$J(\pi_*) - J(\pi_T) \leq \varepsilon_T + O\left(\frac{1}{T} + \frac{1}{T}\sum_{t=0}^{T-1} \varepsilon_t\right), \qquad (16)$$

*where $\pi_* : \mathcal{X} \to \Delta(\mathcal{A})$ is a measurable maximizer of (8).*

Thm. 7 shows that inexact PMD can behave comparably to its exact version provided that 1) the action value functions $\hat{q}_{\pi_t}$ are estimated with increasing accuracy, and 2) that the sequence of policies is $(\mathcal{G}, \mathcal{F})$-compatible, for example in the Sobolev-based setting of Cor. 6. Specifically, if $\|q_{\pi_t} - \hat{q}_{\pi_t}\|_\infty \leq O(1/t)$ for any $t \in \mathbb{N}$, the convergence rate of inexact PMD is of order $O(\log T/T)$, only a logarithmic factor slower than exact PMD. This means that we do not necessarily need a good approximation of the world model from the beginning but rather a strategy to improve upon such approximation as we perform more PMD iterations. This suggests adopting an alternating strategy between exploration (world model learning) and exploitation (PMD steps), as suggested in Sec. 4. We do not investigate this question in this work.

The demonstration technique used to prove Thm. 7 follows closely [12, Thm. 8 and 13]. We provide a proof in Appendix B.1 since the original result did not allow for a decreasing approximation error but rather assumed a constant one. Moreover, extending it to the case of infinite $\mathcal{X}$ requires taking care of additional details related to potential measurability issues.

**Action-value approximation error in terms of World Model estimates**. Thm. 7 highlights the importance of studying the error of the estimator for the action-value functions produced by Alg. 1. These objects are obtained via the formula described in Proposition 3 in terms of $\mathsf{T}_n$, $r_n$ and $\mathsf{P}_\pi$. The exact $q_\pi$ has an analogous closed-form characterization in terms of $\mathsf{T}$, $r$ and $\mathsf{P}_\pi$ as expressed in (7), and motivating our operator-based approach. The following result compares these quantities in terms of the approximation errors of the world model and the reward function.

**Lemma 8** (Implications of the Simulation Lemma). *Let $\mathsf{T}_n$ and $r_n$ the empirical estimators of the transfer operator $\mathsf{T}$ and reward function $r$ as defined in Proposition 3, respectively. If $\mathsf{T}$ satisfies Asm. 1, $r \in \mathcal{G}$, and $\gamma\|\mathsf{T}_n\| < \gamma' < 1$, then, for every $(\mathcal{G}, \mathcal{F})$-compatible policy $\pi$*

$$\|\hat{q}_\pi - q_\pi\|_\infty \leq \frac{1}{1-\gamma'}\left[const_\psi\|r_n - r\|_\mathcal{G} + \frac{\gamma\|r\|_\infty}{1-\gamma}\|\mathsf{T}|_\mathcal{F} - \mathsf{T}_n\|_{\mathsf{HS}}\right].$$

In the result above, when applied to a function in $\mathcal{G}$, such as $r_n$, the uniform norm is to be interpreted as the uniform norm of the evaluation of such function, namely $\|r_n\|_\infty = \sup_{(x,a)\in\Omega} |\langle r_n, \psi(x,a)\rangle|$, and analogously for $\mathsf{T}_n$. The proof, reported in Appendix C.2, follows by first decomposing the difference $q_\pi - \hat{q}_\pi$ with the simulation lemma [31, Lemma 2.2] and then applying the triangular inequality for the uniform norm.

**POWR converges**. We are ready to state the convergence result for Alg. 1. We consider the setting where the dataset used to learn $\mathsf{T}_n$ (and $r_n$) is made of i.i.d. triplets $(x_i, a_i, x_i')$ with $(x_i, a_i)$ sampled from a distribution $\rho \in \mathcal{P}(\Omega)$ supported on all $\Omega$ (such as the state occupancy measure (see e.g. [11] or Appendix A.3) of the uniform policy $\pi(\cdot|x) = 1/|\mathcal{A}|$) and $x_i'$ sampled from $\tau(\cdot|x_i, a_i)$. To guarantee bounds in uniform norm, the result makes a further regularity assumption, of the transfer operator (and the reward function)

**Assumption 2** (Strong Source Condition). *Let $\rho \in \mathcal{P}(\Omega)$ and $C_\rho$ the covariance operator $\sum_{a\in\mathcal{A}} \int_\mathcal{X} \rho(dx, a)\psi(x,a) \otimes \psi(x,a)$. The transition operator $\mathsf{T}$ and the reward function $r$ are such that $\mathsf{T}|_\mathcal{F} \in \mathsf{HS}(\mathcal{F}, \mathcal{G})$ and $r \in \mathcal{G}$. Further, $\left\|(T|_\mathcal{F})^* C_\rho^{-\beta}\right\|_\mathsf{HS} < \infty$ and $\left\|C_\rho^{-\beta} r\right\|_\mathcal{G} < \infty$ for some $\beta > 0$.*

Asm. 2 imposes a strong requirement to the so-called *source condition*, a quantity that describes how well the target objective of the learning process (here $\mathsf{T}$ and $r$) "interact" with the sampling distribution. The assumption is always satisfied when the hypothesis space is finite dimensional (e.g. in the tabular RL setting) and imposes additional smoothness on $\mathsf{T}$ and $r$ when belonging to a Sobolev space. Equipped with this assumption, we can now state the convergence theorem for Alg. 1.

**Theorem 9.** *Let $(\pi_t)_{t\in\mathbb{N}}$ be a sequence of policies generated by Alg. 1 in the same setting of Cor. 6. If the action-value functions $\hat{q}_{\pi_t}$ are estimated from a dataset $(x_i, a_i; x_i')_{i=1}^n$ with $(x_i, a_i) \sim \rho \in \mathcal{P}(\Omega)$ such that Asm. 2 holds with parameter $\beta$, the iterates of Alg. 1 converge to the optimal policy as*

$$J(\pi_*) - J(\pi_T) \leq O\left(\frac{1}{T} + \delta^2 n^{-\alpha}\right)$$

*with probability not less than $1 - 4e^{-\delta}$. Here, $\alpha \in \left(\frac{\beta}{2+2\beta}, \frac{\beta}{1+2\beta}\right)$ and $\pi_* : \mathcal{X} \to \Delta(\mathcal{A})$ is a measurable maximizer of (8).*

The proof of Thm. 9 is reported in Appendix C and combines the results discussed in this section with fast convergence rates for the least-squares [40] and CME [21] estimators. In particular we first use Thm. 5 to guarantee that the policies produced by Alg. 1 are all $(\mathcal{G}, \mathcal{F})$-compatible and therefore that applying Proposition 3 to obtain an estimator for the action-value function is well-defined. Then, we use Lemma 8 to study the approximation error of these estimators in terms of our estimates for the world model and the reward function. Bounds on these quantities are then used in the result for inexact PMD convergence in Thm. 7. We note here that since the latter results require convergence in uniform norm, we cannot leverage standard results for least-squares and CME convergence, which characterize convergence in $L_2(\Omega, \mu)$ and would only require Asm. 1 (Linear MDP) to hold. Rather, we need to impose Asm. 2 to guarantee faster rates in uniform norm.

## 6 Experimental results

We empirically evaluated POWR on classical *Gym* environments [26], ranging from discrete (`FrozenLake-v1`,`Taxi-v3`) to continuous state spaces (`MountainCar-v0`). To ensure balancing between exploration and exploitation of our method, we alternated between running the environment with the current policy to collect samples for world model learning and running Alg. 1 for a number of steps to generate a new policy. Appendix D provides implementation details regarding this process as well as additional results.

Fig. 1 compares our approach with the performance of well-established baselines including A2C [41], DQN [4], TRPO [7], and PPO [6]. The figure reports the average cumulative reward obtained by the models on test environments with respect to the number of interactions with the MDP (*timesteps* in log scale in the figure) across 7 different training runs. In all plots, the horizontal dashed line represents the "success" threshold for the corresponding environment, according to official guidelines. We observe that our method outperforms all competitors by a significant margin in terms of sample complexity, that is, the reward achieved after a given number of timesteps. In the case of the `Taxi-v3`

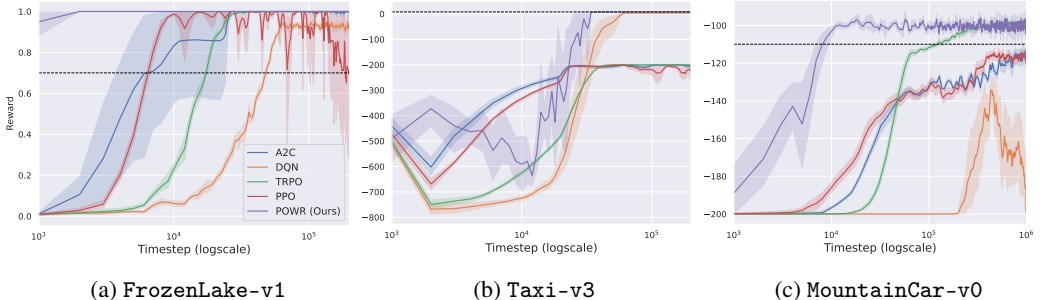

(a) `FrozenLake-v1`          (b) `Taxi-v3`          (c) `MountainCar-v0`

Figure 1: The plots show the average cumulative reward in different environments with respect to the timesteps (i.e. number of interactions with MDP). The dark lines represent the mean of the cumulative reward and the shaded area is the minimum and maximum values reached across 7 independent runs. The horizontal dashed lines represent the reward threshold proposed by the *Gym* library [26].

environment, it avoids converging to a local optimum, in contrast every other method with the exception of DQN. On the downside, we note that our method exhibits less stability than other approaches, particularly during the initial stages of the training process. This is arguably due to a sub-optimal interplay between exploration and exploitation, which will be the subject of future work.

## 7 Conclusions and Future Work

Motivated by recent advancements in policy mirror descent (PMD), this work introduced a novel reinforcement learning (RL) algorithm leveraging these results. Our approach operates in two, possibly alternating, phases: learning a world model and planning via PMD. During exploration, we utilize conditional mean embeddings (CMEs) to learn a world model operator, showing that this procedure is well-posed when performed over suitable Sobolev spaces. The planning phase involves PMD steps for which we guarantee convergence to a global optimum at a polynomial rate under specific MDP regularities.

Our analysis opens avenues for further exploration. Firstly, extending PMD to infinite action spaces remains a challenge. While we introduced the operatorial perspective on RL for infinite state space settings, the PMD update with KL divergence requires approximation methods (e.g., Monte Carlo) whose impact on convergence requires investigation. Secondly, scalability to large environments requires adopting approximated yet efficient CME estimators like Nystrom [35] or reduced-rank regressors [42, 43]. Thirdly, a question we touched upon only empirically, is whether alternating world model learning with inexact PMD updates benefits the exploration-exploitation trade-off. Studying this strategy's impact on convergence is a promising future direction. Finally, a crucial question is generalizing our policy compatibility results beyond Sobolev spaces. Ideally, a representation learning process would identify suitable feature maps that guarantee compatibility with the PMD-generated policies while allowing for added flexibility in learning the world model.

## Acknowledgments and Disclosure of Funding

We acknowledge financial support from NextGenerationEU and MUR PNRR project PE0000013 CUP J53C22003010006 "Future Artificial Intelligence Research (FAIR)", EU grant ELISE (GA no 951847) and EU Project ELIAS (GA no 101120237).

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

# Appendix

The appendices are organized as follows:

- Appendix A discuss the operatorial formulation of RL and show how to derive the operator-based results in this work.
- Appendix B focuses on policy mirror descent (PMD) and its convergence rate in the inexact setting.
- Appendix C proves the main result of this work, namely the theoretical analysis of POWR.
- Appendix D provide details on the experiments reported in this work.

## A Operatorial Results

### A.1 Auxiliary Lemma

We recall here a corollary of the Sherman-Woodbury identity [44].

**Lemma A.1.** *Let $A$ and $B$ two conformable linear operators such that $(I + AB)^{-1}$ is invertible. Then $(I + AB)^{-1}A = A(I + BA)^{-1}$*

*Proof.* The result is obvious if $A$ is invertible. More generally, we consider the following two applications of the Sherman-Woodbury [44] formula

$$(I + AB)^{-1} = I - A(I + BA)^{-1}B \tag{A.1}$$

and

$$(I + BA)^{-1} = I - (I + BA)^{-1}BA. \tag{A.2}$$

Multiplying the two equations by $A$ respectively to the right and to the left, we obtain the desired result. $\qquad\square$

### A.2 Markov operators and their properties

We recall here the notion of Markov operators, which is central for a number of results in the following. We refer to [30, Chapter 19] for more details on the topic.

**Definition A.1** (Markov operators). *Let $\mathcal{X}$ and $\mathcal{Y}$ be Polish spaces. A bounded linear operator $\mathcal{L}(B_b(\mathcal{X}), B_b(\mathcal{Y}))$ is a Markov operator if is positive and maps the unit function to itself, that is:*

> *a. $f \geq 0 \in B_b(\mathcal{X}) \implies \mathsf{P}f \geq 0 \in B_b(\mathcal{Y})$,*
> *b. $\mathsf{P}\mathbf{1}_\mathcal{X} = \mathbf{1}_\mathcal{Y}$,*

*where $\mathbf{1}_\mathcal{X} : \mathcal{X} \to \mathbb{R}$ (respectively $\mathbf{1}_\mathcal{Y}$) denotes the function taking constant value equal to $1$ on $\mathcal{X}$ (respectively $\mathcal{Y}$).*

We recall that Markov operators are a convex subset of $\mathcal{L}(B_b(\mathcal{X}), B_b(\mathcal{Y}))$. Here we denote this space as $\mathcal{L}_\mathrm{M}(B_b(\mathcal{X}), B_b(\mathcal{Y}))$. Direct inspection of (5) and (6) shows that the transfer operator $\mathsf{T}$ associated to an MDP and the policy operator $\mathsf{P}_\pi$ associated to a policy $\pi$ are both Markov operators.

**Markov Operators and Policy Operators**. In (6) we defined the policy operator $\mathsf{P}_\pi$ associated to a policy $\pi$. It turns out that the converse is also true, namely that any such Markov operator is a policy operator.

**Proposition A.2.** *Let $\mathsf{P} \in \mathcal{L}_\mathrm{M}(B_b(\mathcal{X}), B_b(\Omega))$ be a Markov operator. Then there exists $\pi_\mathsf{P}$, such that the associated policy operator corresponds to $\mathsf{P}$, namely $\mathsf{P}_{\pi_\mathsf{P}} = \mathsf{P}$.*

*Proof.* Define the map $\pi_\mathsf{P} : \mathcal{X} \to \mathcal{M}(\mathcal{A})$ taking value in the space of bounded Borel measures over $\mathcal{A}$ such that, for any $x \in \mathcal{X}$ and any $\mathcal{B} \subseteq \mathcal{A}$ Borel measurable subset

$$\pi_\mathsf{P}(\mathcal{B}|x) = (\mathsf{P}\mathbf{1}_{\mathcal{X} \times \mathcal{B}})(x). \tag{A.3}$$

We need to guarantee that for every $x \in \mathcal{X}$ the function $\pi_{\mathsf{P}}(\cdot|x)$ is a signed measure. To show this, first note that the operation defined by $\pi_{\mathsf{P}}$ is well-defined, since for any measurable set $\mathcal{B}$ the function $\mathbf{1}_{\mathcal{X} \times \mathcal{B}}$ is also measurable, making $\pi_{\mathsf{P}}(\mathcal{B}|x)$ well defined as well. Moreover, since $\mathbf{1}_{\emptyset}(a) = 0$ for any $a \in \mathcal{A}$, it implies that $\mathbf{1}_{\mathcal{X} \times \emptyset} = 0$ and therefore $\pi_{\mathsf{P}}(\emptyset|x) = 0$ for any $x \in \mathcal{X}$. Finally, $\sigma$-additivity follows from the definition of indicator functions, namely $\mathbf{1}_{\bigcup_{i=1}^{\infty} \mathcal{B}_i} = \sum_{i=1}^{\infty} \mathbf{1}_{\mathcal{B}_i}$ for any family of pair-wise disjoint sets $(\mathcal{B}_i)_{i=1}^{n}$, which implies $\pi_{\mathsf{P}}\left(\bigcup_{i=1}^{\infty} \mathcal{B}_i|x\right) = \sum_{i=1}^{\infty} \pi_{\mathsf{P}}(\mathcal{B}_i|x)$ for any $x \in \mathcal{X}$.

We now apply the two properties of Markov operators to show that $\pi_{\mathsf{P}}$ takes values in $\mathcal{P}(\mathcal{A})$, namely it is a non-negative measure that sums to $1$. Since Markov operators map non-negative functions in non-negative functions and since $\mathbf{1}_{\mathcal{X} \times \mathcal{B}} \geq 0$ for any $\mathcal{B} \subseteq \mathcal{X}$, we have $\pi(\cdot|x) \geq 0$ as well for any $x \in \mathcal{X}$. Moreover, since $\Omega = \mathcal{X} \times \mathcal{A}$ and $\mathsf{P}\mathbf{1}_{\Omega} = \mathbf{1}_{\mathcal{X}}$, we have

$$\pi(\mathcal{A}|x) = (\mathsf{P}\,\mathbf{1}_{\Omega})(x) = \mathbf{1}_{\mathcal{X}}(x) = 1, \tag{A.4}$$

for any $x \in \mathcal{X}$. Therefore $\pi_{\mathsf{P}}(\cdot|x)$ is a probability measure for any $x \in \mathcal{X}$. Direct application of (6) shows that the associated policy operator corresponds to $\mathsf{P}$, namely $\mathsf{P}_{\pi_{\mathsf{P}}} = \mathsf{P}$ as desired. $\qquad\square$

Given the correspondence between policies and their Markov operator according to (6) and Proposition A.2, in the following we will denote the policy operator associated to a policy $\pi$ only $\mathsf{P}$ where clear from context.

With the definition of the Markov operator in place, we can now prove the following result introduced in the main paper.

**Proposition 2** (Well-specified CME). *Under Assumption 1, $(\mathsf{T}|_{\mathcal{F}})^* = (\mathsf{T}^*)|_{\mathcal{G}}$ and, for any $(x, a) \in \Omega$*

$$(\mathsf{T}|_{\mathcal{F}})^* \psi(x, a) = \int_{\mathcal{X}} \varphi(x')\,\tau(x'|x, a) = \mathbb{E}[\varphi(X')|X = x, A = a]. \tag{9}$$

*Proof.* Recall that since they are Hilbert spaces $\mathcal{F} \cong \mathcal{F}^*$ and $\mathcal{G} \cong \mathcal{G}^*$ are isometric to their dual and therefore we can interpret any $f \in \mathcal{F}$ as the function $f(\cdot) = \langle f, \varphi(\cdot) \rangle$ with some abuse of notation, where clear from context. By Assumption 1 we have that $\mathsf{T}|_{\mathcal{F}}$ takes values in $\mathcal{G}$. This means that $(\mathsf{T}|_{\mathcal{F}} f) \in \mathcal{G}$ or, in other words

$$
\begin{aligned}
\langle \mathsf{T}|_{\mathcal{F}} f, \psi(x, a) \rangle &= (\mathsf{T}|_{\mathcal{F}} f)(x, a) \\
&= \int_{\mathcal{F}} f(x')\,\tau(dx'|x, a) \\
&= \int_{\mathcal{F}} \langle f, \varphi(x') \rangle\,\tau(dx'|x, a) \\
&= \left\langle f, \int \varphi(x')\,\tau(dx'|x, a) \right\rangle,
\end{aligned}
$$

from which we obtain

$$\langle f, (\mathsf{T}|_{\mathcal{F}})^* \psi(x, a) \rangle = \left\langle f, \int \varphi(x')\,\tau(dx'|x, a) \right\rangle.$$

Since the above equality holds for any $f \in \mathcal{F}$ (9) holds, as desired. $\qquad\square$

We note that the result can be extended to the setting where $\mathsf{T}|_{\mathcal{F}}(\mathcal{F}) \subseteq \mathcal{G}$, namely the image of $\mathsf{T}|_{\mathcal{F}}$ is contained in $\mathcal{G}$, namely a sort of $(\mathcal{F}, \mathcal{G})$-compatibility for the transition operator (see Definition 1).

## A.3 The operatorial formulation of RL

According to the operatorial characterization in (7), the action value function of a policy $\pi$ is directly related to the action of the associated policy operator $\mathsf{P}$. To highlight this relation, we will adopt the following notation:

- **Action-value (or Q-)function.**

$$q(\mathsf{P}) = (\mathsf{Id} - \gamma \mathsf{T}\mathsf{P})^{-1}\, r. \tag{A.5}$$

- **Value function.**

$$v(\mathsf{P}) = \mathsf{P}q(\mathsf{P}). \tag{A.6}$$

- **Cumulative reward.** The RL objective functional

$$J(\mathsf{P}) = \left\langle \mathsf{P}\left(\mathsf{Id} - \gamma\mathsf{TP}\right)^{-1} r, \nu \right\rangle = \langle \mathsf{P}q(\mathsf{P}), \nu \rangle = \langle v(\mathsf{P}), \nu \rangle. \tag{A.7}$$

- **State visitation (or State occupancy) measure.** By the characterization of the adjoints of $\mathsf{P}$ and $\mathsf{T}$ (see discussion in Section 3 we can represent the evolution of a state distribution $\nu_t$ at time $t$ to the next state distribution as $\nu_{t+1} = \mathsf{T}^*\mathsf{P}^*\nu_t$. Applying this relation recursively, we recover the *state visitation probability* associated to the starting state distribution $\nu_0 = \nu \in \mathcal{P}(\mathcal{X})$, the MDP with transition $\mathsf{T}$ and the policy $\mathsf{P}$ as

$$d_\nu(\mathsf{P}) = (1 - \gamma)\sum_{t=0}^{\infty} \gamma^t (\mathsf{T}^*\mathsf{P}^*)^t \nu = (1 - \gamma)\left(\mathsf{Id} - \gamma\mathsf{PT}\right)^{-*} \nu, \tag{A.8}$$

where the $(1 - \gamma)\gamma^t$ is a normalizing factor to guarantee that the series corresponds to a convex combination of the probability distributions $\nu_t$, hence guaranteeing $d_\nu(\mathsf{P})$ to be well-defined (namely it belongs to $\mathcal{P}(\mathcal{X})$).

**Previous well-known RL results in operator form.** Under the operatorial formulation of RL, we can recover several well-known results from the reinforcement literature with concise proofs. We recall here a few of these results that will be useful in the following.

**Remark A.1.** *Algebraic manipulation of the cumulative expected reward $J(\mathsf{P})$ implies*

$$J(\mathsf{P}) = \left\langle \mathsf{P}\left(\mathsf{Id} - \gamma\mathsf{TP}\right)^{-1} r, \nu \right\rangle = \left\langle \mathsf{P}r, \left(\mathsf{Id} - \gamma\mathsf{PT}\right)^{-*} \nu \right\rangle = \frac{1}{1 - \gamma} \langle \mathsf{P}r, d_\nu(\mathsf{P}) \rangle,$$

*where we used Lemma A.1 and $d_\nu(\mathsf{P})$ is the state visitation distribution starting from $\nu$ and following the policy $\mathsf{P}$.*

The following result, known as *Performance Difference* Lemma [see e.g. 11, Lemma 1.16], will be instrumental to prove the convergence rates for PMD in Theorem 7.

**Lemma A.3** (Performance difference). *Let $\mathsf{P}_1$, $\mathsf{P}_2$ two policy operators. The following equality holds*

$$J(\mathsf{P}_1) - J(\mathsf{P}_2) = \frac{1}{1 - \gamma} \langle (\mathsf{P}_1 - \mathsf{P}_2)q(\mathsf{P}_2), d_\nu(\mathsf{P}_1) \rangle. \tag{A.9}$$

*Proof.* Using the definition of $J(\mathsf{P}_1)$ and Lemma A.1 one gets

$$\begin{aligned}
J(\mathsf{P}_1) - J(\mathsf{P}_2) &= \left\langle \mathsf{P}_1\left(\mathsf{Id} - \gamma\mathsf{TP}_1\right)^{-1} r, \nu \right\rangle - \left\langle \mathsf{P}_2\left(\mathsf{Id} - \gamma\mathsf{TP}_2\right)^{-1} r, \nu \right\rangle \\
&= \left\langle \left(\mathsf{Id} - \gamma\mathsf{P}_1\mathsf{T}\right)^{-1} \mathsf{P}_1 r, \nu \right\rangle - \left\langle \mathsf{P}_2\left(\mathsf{Id} - \gamma\mathsf{TP}_2\right)^{-1} r, \nu \right\rangle \\
&= \left\langle \left(\mathsf{Id} - \gamma\mathsf{P}_1\mathsf{T}\right)^{-1} \mathsf{P}_1\left(\mathsf{Id} - \gamma\mathsf{TP}_2\right)\left(\mathsf{Id} - \gamma\mathsf{TP}_2\right)^{-1} r, \nu \right\rangle \\
&\quad - \left\langle \left(\mathsf{Id} - \gamma\mathsf{P}_1\mathsf{T}\right)^{-1}\left(\mathsf{Id} - \gamma\mathsf{P}_1\mathsf{T}\right)\mathsf{P}_2\left(\mathsf{Id} - \gamma\mathsf{TP}_2\right)^{-1} r, \nu \right\rangle \\
&= \left\langle \left(\mathsf{Id} - \gamma\mathsf{P}_1\mathsf{T}\right)^{-1}\left[\mathsf{P}_1\left(\mathsf{Id} - \gamma\mathsf{TP}_2\right) - \left(\mathsf{Id} - \gamma\mathsf{P}_1\mathsf{T}\right)\mathsf{P}_2\right]\left(\mathsf{Id} - \gamma\mathsf{TP}_2\right)^{-1} r, \nu \right\rangle \\
&= \left\langle \left(\mathsf{Id} - \gamma\mathsf{P}_1\mathsf{T}\right)^{-1}\left[\mathsf{P}_1 - \mathsf{P}_2\right]\left(\mathsf{Id} - \gamma\mathsf{TP}_2\right)^{-1} r, \nu \right\rangle \\
&= \frac{1}{1 - \gamma} \langle (\mathsf{P}_1 - \mathsf{P}_2)q(\mathsf{P}_2), d_\nu(\mathsf{P}_1) \rangle.
\end{aligned}$$

$\square$

A direct consequence of the operator formulation of the performance difference lemma is the following operator-based characterization of the differential behavior of the RL objective. The result can be found in [11] for the case of finite state and action spaces, however here the operatorial formulation allows for a much more concise proof.

**Corollary A.4** (Directional derivatives). *For any two Markov* $\mathsf{P}_1, \mathsf{P}_2 : B_b(\mathcal{X}) \to B_b(\Omega)$, *we have that the directional derivative in* $\mathsf{P}_1$ *towards* $\mathsf{P}_2$ *is*

$$\lim_{h \to 0} \frac{J(\mathsf{P}_1 + h(\mathsf{P}_2 - \mathsf{P}_1)) - J(\mathsf{P}_1)}{h} = \frac{1}{1 - \gamma} \langle (\mathsf{P}_2 - \mathsf{P}_1) q(\mathsf{P}_1), d_\nu(\mathsf{P}_1) \rangle. \qquad (A.10)$$

*Proof.* The result follows by recalling that the space of Markov operators is convex, namely for any $h \in [0, 1]$ the term $\mathsf{P}_h = \mathsf{P}_1 + h(\mathsf{P}_2 - \mathsf{P}_1)$ is still a Markov operator. Therefore, we can apply Lemma A.3 to obtain

$$J(\mathsf{P}_h) - J(\mathsf{P}_1) = \langle (\mathsf{P}_h - \mathsf{P}_1) q(\mathsf{P}_1), d_\nu(\mathsf{P}_h) \rangle \qquad (A.11)$$

$$= h \langle (\mathsf{P}_2 - \mathsf{P}_1) q(\mathsf{P}_1), d_\nu(\mathsf{P}_h) \rangle. \qquad (A.12)$$

We can therefore divide the above quantity by $h$ and send $h \to 0$. The result follows by observing that $d_\nu(\mathsf{P}_h) = (\mathsf{Id} - \gamma \mathsf{T} \mathsf{P}_h)^{-*} \nu \to (\mathsf{Id} - \gamma \mathsf{T} \mathsf{P}_1)^{-*} \nu = d_\nu(\mathsf{P}_1)$ for $h \to 0$, since $\|\mathsf{P}_h\| = 1$ for any $h \in [0, 1]$ and the function $\mathsf{M} \mapsto (I - \gamma \mathsf{T} \mathsf{M})^{-1}$ is continuous on the open ball of radius $1/\gamma > 1$ in $\mathcal{L}(B_b(\Omega), B_b(\mathcal{X}))$ with respect to the operator norm. $\qquad \square$

**Properties of** $(\mathsf{Id} - \gamma \mathsf{P} \mathsf{T})^{-1}$. The quantity $(\mathsf{Id} - \gamma \mathsf{P} \mathsf{T})^{-1}$ (note, not $(\mathsf{Id} - \gamma \mathsf{T} \mathsf{P})^{-1}$) plays a central role in the study of POWR. We prove here a few properties that will be useful in the following.

**Lemma A.5** (Properties of $\mathsf{Id} - \gamma \mathsf{P} \mathsf{T}$). *The following facts are true:*

1. *For any* $f \geq 0 \in B_b(\mathcal{X})$ *it holds* $(\mathsf{Id} - \gamma \mathsf{P} \mathsf{T})^{-1} f \geq f$.

2. *The operator* $(1 - \gamma)(\mathsf{Id} - \gamma \mathsf{P} \mathsf{T})^{-1}$ *is a Markov operator.*

3. *For any positive measure* $\nu \in \mathcal{M}(B_b(\mathcal{X}))$ *it holds* $(1 - \gamma) \|(\mathsf{Id} - \gamma \mathsf{P} \mathsf{T})^{-*} \nu\|_{\mathrm{TV}} = \|\nu\|_{\mathrm{TV}}$.

4. *For any positive measure* $\nu \in \mathcal{M}(B_b(\mathcal{X}))$ *it holds* $\|\mathsf{P}^* \nu\|_{\mathrm{TV}} = \|\nu\|_{\mathrm{TV}}$.

5. *For any bounded linear operator* $\mathsf{X}$, *policy operator* $\mathsf{P}$ *and discount factor* $\gamma < \|\mathsf{X}\|$, *it holds* $\|(\mathsf{Id} - \gamma \mathsf{X} \mathsf{P})^{-1}\|_\infty \leq 1/(1 - \gamma \|\mathsf{X}\|)$.

*Proof.* Since both $\mathsf{T}$ and $\mathsf{P}$ are Markov operators by construction, it immediately follows that their composition is a Markov operator as well. Using the Neumann series representation of $(\mathsf{Id} - \gamma \mathsf{P} \mathsf{T})^{-1}$ it follows that for all $f \geq 0 \in B_b(\mathcal{X})$

$$(\mathsf{Id} - \gamma \mathsf{P} \mathsf{T})^{-1} f = \sum_{t=0}^\infty \gamma^t (\mathsf{P} \mathsf{T})^t f = f + \sum_{t=1}^\infty \gamma^t (\mathsf{P} \mathsf{T})^t f \geq f \geq 0,$$

proving (1). Further,

$$(\mathsf{Id} - \gamma \mathsf{P} \mathsf{T})^{-1} \mathbf{1}_\mathcal{X} = \sum_{t=0}^\infty \gamma^t (\mathsf{P} \mathsf{T})^t \mathbf{1}_\mathcal{X} = \sum_{t=0}^\infty \gamma^t \mathbf{1}_\mathcal{X} = \frac{\mathbf{1}_\mathcal{X}}{1 - \gamma}$$

showing that $(1 - \gamma)(\mathsf{Id} - \gamma \mathsf{P} \mathsf{T})^{-1} \mathbf{1}_\mathcal{X} = \mathbf{1}_\mathcal{X}$ and proving (2). Finally, since $(1 - \gamma)(\mathsf{Id} - \gamma \mathsf{P} \mathsf{T})^{-1}$ and $\mathsf{P}$ are Markov operators, (3) and (4) follow from the direct application of [30, Theorem 19.2]. For the last point (5), let $f \in B_b(\Omega)$. As $\mathsf{P}$ is a conditional expectation operator, it holds that

$$\|\mathsf{X} \mathsf{P} f\|_\infty \leq \|\mathsf{X}\| \, \mathsf{P}(\mathbf{1}_\Omega \|f\|_\infty) = \|\mathsf{X}\| \, \|f\|_\infty$$

Where the inequality is just the conditional version of Jensen's inequality [45, Chapter 5] applied on the (convex) $\|\cdot\|_\infty$ function, while the equality comes from the fact that $\mathsf{T} \mathsf{P}$ is a Markov operator. Then, we have

$$\sup_{\|f\|_\infty = 1} \|(\mathsf{Id} - \gamma \mathsf{X} \mathsf{P})^{-1} f\|_\infty = \sup_{\|f\|_\infty = 1} \|\sum_{t=0}^\infty (\gamma \mathsf{X} \mathsf{P})^t f\|_\infty$$

$$\leq \sup_{\|f\|_\infty = 1} \sum_{t=0}^\infty \gamma^t \|(\mathsf{X} \mathsf{P})^t f\|_\infty$$

$$\leq \sup_{\|f\|_\infty = 1} \sum_{t=0}^\infty \gamma^t \|\mathsf{X}\|^t \|f\|_\infty$$

$$(\|f\|_\infty = 1) = \frac{1}{1 - \gamma \|\mathsf{X}\|}.$$

$\square$

**Simulation Lemma.** We report here the Simulation lemma, since it will be key to bridging the gap between Policy Mirror Descent and Conditional Mean Embeddings in Theorem 9 through Lemma 8.

**Lemma A.6** (Simulation Lemma [31]-Lemma 2.2). *Let $\gamma > 0$ and let $\mathsf{T}_1$, $\mathsf{T}_2$ two linear operators with operator norm strictly less than $\gamma$. Let $\mathsf{P}$ be a policy operator. Denote by $q(\mathsf{P}, \mathsf{T}) = (\mathsf{Id} - \gamma\mathsf{TP})^{-1}r$ the (generalized) action-value function associated to these terms and $v(\mathsf{P}, \mathsf{T}) = \mathsf{P}q(\mathsf{P}, \mathsf{T})$ the corresponding value function. Then the following equality holds*

$$q(\mathsf{P}, \mathsf{T}_1) - q(\mathsf{P}, \mathsf{T}_2) = \gamma\left(\mathsf{Id} - \gamma\mathsf{T}_1\mathsf{P}\right)^{-1}\left(\mathsf{T}_2 - \mathsf{T}_1\right)v(\mathsf{P}, \mathsf{T}_2) \tag{A.13}$$

*Proof.* Using the same technique of the proof of Lemma A.3 one has

$$\begin{aligned}
q(\mathsf{P}, \mathsf{T}_1) - q(\mathsf{P}, \mathsf{T}_2) &= \left(\mathsf{Id} - \gamma\mathsf{T}_1\mathsf{P}\right)^{-1}r - \left(\mathsf{Id} - \gamma\mathsf{T}_2\mathsf{P}\right)^{-1}r \\
&= \gamma\left(\mathsf{Id} - \gamma\mathsf{T}_1\mathsf{P}\right)^{-1}\left(\mathsf{T}_2 - \mathsf{T}_1\right)\mathsf{P}\left(\mathsf{Id} - \gamma\mathsf{T}_2A\right)^{-1}r \\
&= \gamma\left(\mathsf{Id} - \gamma\mathsf{T}_1\mathsf{P}\right)^{-1}\left(\mathsf{T}_2 - \mathsf{T}_1\right)v(\mathsf{P}, \mathsf{T}_2)
\end{aligned}$$

where we have used fact that for any two invertible operators $\mathsf{M}$ and $\mathsf{P}$ it holds $\mathsf{M}^{-1} - \mathsf{P}^{-1} = \mathsf{M}^{-1}(\mathsf{P} - \mathsf{M})\mathsf{P}^{-1}$ for the second equation and applied the operatorial characterization of the value function to conclude the proof. $\square$

We then have the following result, which hinges on a generalization of the standard Simulation lemma in [31, Lemma 2.2] where we account also for the reward function to vary.

**Corollary A.7.** *Let $\gamma > 0$ and let $\mathsf{T}_1$, $\mathsf{T}_2$ two linear operators with operator norm strictly less than $\gamma$. Let $r_1$ and $r_2$ be two reward functions and $\mathsf{P}$ a policy operator. Denote by $q(\mathsf{P}, \mathsf{T}, r) = (\mathsf{Id} - \gamma\mathsf{TP})^{-1}r$ the (generalized) action-value function associated to these terms and $v(\mathsf{P}, \mathsf{T}, r) = \mathsf{P}q(\mathsf{P}, \mathsf{T}, r)$ the corresponding value function. Then the following equality holds*

$$q(\mathsf{P}, \mathsf{T}_1, r_1) - q(\mathsf{P}, \mathsf{T}_2, r_2) = (\mathsf{Id} - \gamma\mathsf{T}_1\mathsf{P})^{-1}(r_1 - r_2) + \gamma(\mathsf{Id} - \gamma\mathsf{T}_1\mathsf{P})^{-1}(\mathsf{T}_2 - \mathsf{T}_1)v(\mathsf{P}, \mathsf{T}_2, r_2).$$

*Proof.* The difference between action-value functions can be written as

$$\begin{aligned}
q(\mathsf{P}, \mathsf{T}_1, r_1) - q(\mathsf{P}, \mathsf{T}_2, r_2) &= (\mathsf{Id} - \gamma\mathsf{T}_1\mathsf{P})^{-1}r_1 - (\mathsf{Id} - \gamma\mathsf{T}_2\mathsf{P})^{-1}r_2 \\
&= (\mathsf{Id} - \gamma\mathsf{T}_1\mathsf{P})^{-1}(r_1 - r_2) + \left[(\mathsf{Id} - \gamma\mathsf{T}_1\mathsf{P})^{-1} - (\mathsf{Id} - \gamma\mathsf{T}_2\mathsf{P})^{-1}\right]r_2
\end{aligned}$$

where we added and removed a term $(\mathsf{Id} - \gamma\mathsf{T}_1\mathsf{P})^{-1}r_2$. The result follows by plugging in the Simulation Lemma A.6 for the second term of the right hand side. $\square$

The corollary above will be useful in Appendix C to control the approximation error of the estimates $\hat{q}_{\pi_t}$ appearing in the convergence rates for inexact PMD in Theorem 7.

### A.4 Action-value Estimator for $(\mathcal{G}, \mathcal{F})$-compatible Policies

We can leverage the notation introduced in this section to prove the following form for the world model-based estimator of the action-value function.

**Proposition 3.** *Let $\mathsf{T}_n = S_n^*BZ_n \in \mathsf{HS}(\mathcal{F}, \mathcal{G})$ and $r_n = S_n^*b \in \mathcal{G}$ for respectively a $B \in \mathbb{R}^{n \times n}$ and $b \in \mathbb{R}^n$. Let $\pi$ be $(\mathcal{G}, \mathcal{F})$-compatible. Then,*

$$\hat{q}_\pi = (\mathsf{Id} - \gamma\mathsf{T}_n\mathsf{P}_\pi)^{-1}r_n = S_n^*(\mathsf{Id} - \gamma BM_\pi)^{-1}b \tag{12}$$

*where $M_\pi = Z_n\mathsf{P}_\pi S_n^* \in \mathbb{R}^{n \times n}$ is the matrix with entries*

$$\left(M_\pi\right)_{ij} = \langle\varphi(x_i'), \mathsf{P}_\pi\psi(x_j, a_j)\rangle = \int_\mathcal{A}\langle\psi(x_i', a), \psi(x_j, a_j)\rangle\ \pi(da|x_i'). \tag{13}$$

*Proof.* By hypothesis

$$\hat{q}_\pi = (\mathsf{Id} - \gamma\mathsf{T}_n\mathsf{P}_\pi)^{-1}r_n = (\mathsf{Id} - \gamma S_n^*BZ_n\mathsf{P}_\pi)^{-1}S_n^*b. \tag{A.14}$$

Eq. (12) follows by applying Lemma A.1. Eq. (13) can be verified by direct calculation. Denote by $(e_i)_{i=1}^m$ the vectors of the canonical basis in $\mathbb{R}^n$. Then, for any $i, j = 1, \ldots, n$

$$(M_\pi)_{ij} = \langle e_i, M_\pi e_j \rangle = \langle e_i, Z_n \mathsf{P}_\pi S_n^* e_j \rangle = \langle Z_n^* e_i, \mathsf{P}_\pi S_n^* e_j \rangle . \tag{A.15}$$

Now, we recall that the two operators $S_n : \mathcal{G} \to \mathbb{R}^n$ and $Z_n : \mathcal{F} \to \mathbb{R}^n$ are the evaluation operators for respectively the points $(x_i, a_i)_{i=1}^n$ and $(x_i')_{i=1}^n$. Namely, for any vector $v \in \mathbb{R}^n$

$$S_n^* v = \sum_{i=1}^n v_i \psi(x_i, a_i) \qquad \text{and} \qquad Z_n^* v = \sum_{i=1}^n v_i \varphi(x_i'). \tag{A.16}$$

This implies that

$$(M_\pi)_{ij} = \langle Z_n^* e_i, \mathsf{P}_\pi S_n^* e_j \rangle = \langle \varphi(x_i'), \mathsf{P}_\pi \psi(x_j, a_j) \rangle \tag{A.17}$$

Since $\mathsf{P}_\pi$ is $(\mathcal{G}, \mathcal{F})$-compatible by hypothesis, we can leverage the same reasoning used in Proposition 2 to show that

$$(\mathsf{P}_\pi|_{\mathcal{G}})^* \varphi(x') = \int_{\mathcal{A}} \psi(x', a) \, \pi(da|x') \tag{A.18}$$

for any $x' \in \mathcal{X}$. By plugging this equation in the previous characterization for $(M_\pi)_{ij}$ we have

$$\langle \varphi(x_i'), \mathsf{P}_\pi \psi(x_j, a_j) \rangle = \langle \mathsf{P}_\pi^* \varphi(x_i'), \psi(x_j, a_j) \rangle \tag{A.19}$$

$$= \left\langle \int_{\mathcal{A}} \psi(x_i', a) \, \pi(da|x_i'), \psi(x_j, a_j) \right\rangle \tag{A.20}$$

$$= \int_{\mathcal{A}} \langle \psi(x_i', a), \psi(x_j, a_j) \rangle \, \pi(da|x_i'), \tag{A.21}$$

as required. $\qquad \square$

### A.5 Separable Spaces

We show here the sufficient condition for $(\mathcal{G}, \mathcal{F})$-compatibility of a policy in the case of the separable spaces introduced in Section 4.

**Proposition 4** (Separable Spaces). *Let $\phi : \mathcal{X} \to \mathcal{H}$ be a feature map into a Hilbert space $\mathcal{H}$. Let $\mathcal{F} = \mathcal{H} \otimes \mathcal{H}$ and $\mathcal{G} = \mathbb{R}^{|\mathcal{A}|} \otimes \mathcal{H}$ with feature maps respectively $\varphi(x) = \phi(x) \otimes \phi(x)$ and $\psi(x, a) = \phi(x) \otimes e_a$, with $e_a \in \mathbb{R}^{|\mathcal{A}|}$ the one-hot encoding of action $a \in \mathcal{A}$. Let $\pi : \mathcal{X} \to \Delta(\mathcal{A})$ be a policy such that $\pi(a|\cdot) = \langle p_a, \phi(\cdot) \rangle$ with $p_a \in \mathcal{H}$ for any $a \in \mathcal{A}$. Then, $\pi$ is $(\mathcal{G}, \mathcal{F})$-compatible.*

*Proof.* The proposition follows from observing that for any $v \in \mathbb{R}^{|\mathcal{A}|}$ and $h \in \mathcal{H}$, applying $\mathsf{P}_\pi$ according to (6) to the function $g(x, a) = \langle h, \phi(x) \rangle \langle v, De_a \rangle$ yields

$$(\mathsf{P}_\pi g)(x) = \sum_{a \in \mathcal{A}} g(x, a) \pi(a|x) = \langle h, \phi(x) \rangle \sum_{a \in \mathcal{A}} \langle v, De_a \rangle \pi(a|x) \tag{A.22}$$

$$= \langle h, \phi(x) \rangle \sum_{a \in \mathcal{A}} \langle v, De_a \rangle \langle p_a, \phi(x) \rangle \tag{A.23}$$

$$= \left\langle h \otimes \sum_{a \in \mathcal{A}} \langle v, De_a \rangle p_a, \phi(x) \otimes \phi(x) \right\rangle \tag{A.24}$$

Hence $(\mathsf{P}_\pi g)(x) = \langle f, \varphi(x) \rangle$ with $f = h \otimes h' \in \mathcal{H} \otimes \mathcal{H} = \mathcal{F}$ and $h' = \sum_{a \in \mathcal{A}} \langle v, De_a \rangle p_a \in \mathcal{H}$. Therefore, the restriction of $\mathsf{P}_\pi$ to $\mathcal{G}$ takes value in $\mathcal{F}$ as desired. $\qquad \square$

## B   Policy Mirror Descent

In this section we briefly review the tools needed to formulate the PMD method and discuss the convergence rates for inexact PMD. Most of the discussion follows the presentation in [12] formulated within the notation used in this work.

Let $D : \Delta(\mathcal{A}) \times \mathrm{rint}\Delta(\mathcal{A}) \to \mathbb{R}$ a Bregman divergence [36, Definition 9.2] over the probability simplex, where $\mathrm{rint}\Delta(\mathcal{A})$ denotes the relative interior of $\Delta(\mathcal{A})$. In the following, for any $t \in \mathbb{N}$ we will denote by $\pi_t$ the policy produced at iteration $t$ by a PMD algorithm according to the update (2) (with either the exact action-value function or an estimator, as discussed in Section 3) with divergence $D$ and step-size $\eta > 0$. We denote $\mathsf{P}_t = \mathsf{P}_{\pi_t}$ the associated operator. We recall here the PMD update from (2), highlighting the dependency on the policy operator $\mathsf{P}_t$ via the action-value function $q(\mathsf{P}_t)$.

$$\pi_{t+1}(\cdot|x) \in \operatorname*{argmin}_{p \in \Delta(\mathcal{A})} \left\{ -\eta \sum_{a \in \mathcal{A}} q(\mathsf{P}_t)(x, a)p_a + D(p; \pi_t(\cdot|x)) \right\} \quad \text{for all } x \in \mathcal{X}. \tag{B.1}$$

While this point-wise characterization is sufficient to define the updated policy $\pi_{t+1} : \mathcal{X} \to \Delta(\mathcal{A})$ from the previous $\pi_t$ and its action-value function $q(\mathsf{P}_t)$, we need to guarantee that $\pi_{t+1}$ is measurable. If that were not the case, we would not be able to guarantee the existence of a $\mathsf{P}_{t+1}$ associated with it, possibly affecting the well-definiteness of iteratively applying the mirror descent update (B.1). The following result addresses this issue.

**Lemma B.1** (Measurability of the Mirror Descent updates). *Let $D : \Delta(\mathcal{A}) \times \mathrm{rint}\,\Delta(\mathcal{A}) \to \mathbb{R}$ be a Bregman divergence continuous in its first argument. There exists a measurable policy $\pi_{t+1} : \mathcal{X} \to \Delta(\mathcal{A})$ that satisfies (B.1) for all $x \in \mathcal{X}$.*

*Proof.* The proof follows from the Measurable Maximum Theorem [30, Theorem 18.19]. Let us denote $f_t : \mathcal{X} \times \Delta(\mathcal{A}) \to \mathbb{R}$ the function

$$f_t(x, p) := -\eta \sum_{a \in \mathcal{A}} q(\mathsf{P}_t)(x, a)p_a + D(p; \pi_t(x)). \tag{B.2}$$

Let also $\kappa : \mathcal{X} \twoheadrightarrow \Delta(\mathcal{A})$ be the constant correspondance $x \mapsto \Delta(\mathcal{A})$ for all $x \in \mathcal{X}$. $\kappa$ clearly has nonempty compact values, and it is also weakly measurable since for any open set $G \subset \Delta(\mathcal{A})$, its lower inverse $\kappa^\ell(G) := \{x \in \mathcal{X} : \kappa(x) \cap G \neq \varnothing\} = \mathcal{X}$ belongs to the Borel sigma-algebra of $\mathcal{X}$. Finally, since $q(\mathsf{P}_t) \in B_b(\Omega)$, and by assumption $D$ is continuous in its first argument, then we have that $f_t$ is a Carathéodory function. Then, by [30, Theorem 18.19] we have that the correspondence of minimizers $\mu : \mathcal{X} \twoheadrightarrow \Delta(\mathcal{A})$ defined as

$$\mu(x) := \left\{ p_* \in \kappa(x) : f_t(x, p_*) = \min_{p \in \kappa(x)} f_t(x, p) \right\}$$

admits a measurable selector, which we denote $\pi_{t+1} : \mathcal{X} \to \Delta(\mathcal{A})$, proving the statement of the Lemma. $\qquad\square$

The previous Lemma is the key technical step enabling us to extend the convergence rates of Mirror Descent proved in [12] to non-tabular settings. We now state and prove fa ew Lemmas instrumental to prove Theorem 7.

**Lemma B.2** (Three-points lemma). *Let $\pi_{t+1} : \mathcal{X} \to \Delta(\mathcal{A})$ a measurable minimizer of (B.2) and $\mathsf{P}_{t+1}$ its associated operator. For every measurable policy $\pi : \mathcal{X} \to \Delta(\mathcal{A})$ (alongside its associated operator $\mathsf{P}$) it holds*

$$\eta \left[ (\mathsf{P}_{t+1} - \mathsf{P})q(\mathsf{P}_t) \right](x) \geq D(\pi(x); \pi_{t+1}(x)) - D(\pi(x); \pi_t(x)) \tag{B.3}$$

*Proof.* The function $f_t(x, p)$ in (B.2) is convex and differentiable in $p$ as it is a sum of a linear function and a (strictly convex) Bregman divergence. By the first-order optimality condition [36, Corollay 3.68], a minimizer $p_*$ of $f_t(x, \cdot)$ satisfies, for all $p \in \Delta(\mathcal{A})$

$$\langle \nabla f_t(x, p_*), \pi(x) - p_* \rangle \geq 0. \tag{B.4}$$

Since $\pi_{t+1}(x)$ is a minimizer of $f_t(x, \cdot)$ by assumption, letting $p_* = \pi_{t+1}(x)$, the first order optimality condition (B.4) becomes

$$\eta \left[ (\mathsf{P}_{t+1} - \mathsf{P})q(\mathsf{P}_t) \right](x) - \langle \nabla \psi(\pi_t(x)) - \nabla \psi(\pi_{t+1}(x)), \pi(x) - \pi_{t+1}(x) \rangle \geq 0 \quad \Longrightarrow$$

$$\eta \left[ (\mathsf{P}_{t+1} - \mathsf{P})q(\mathsf{P}_t) \right](x) \geq D(\pi(x); \pi_{t+1}(x)) - D(\pi(x); \pi_t(x)) + D(\pi_{t+1}(x); \pi_t(x)) \quad \Longrightarrow$$

$$\eta \left[ (\mathsf{P}_{t+1} - \mathsf{P})q(\mathsf{P}_t) \right](x) \geq D(\pi(x); \pi_{t+1}(x)) - D(\pi(x); \pi_t(x))$$

Where in the first line we used the definition of Bregman divergence [36, Definition 9.2] $D(p; q) := \psi(p) - \psi(q) - \langle \nabla \psi(q), p - q \rangle$ for a suitable Legendre function $\psi : \Delta(\mathcal{A}) \to \mathbb{R}$, the first implication follows from the three-points property of Bregman divergences [36, Lemma 9.11], and the last implication from the positivity of $D(\pi_{t+1}(x); \pi_t(x))$. □

**Corollary B.3** (MD Iterations are monotonically increasing). *This Corollary is essentially a restatement of [12, Lemma 7]. Let $(\mathsf{P}_t)_{t \in \mathbb{N}}$ be the sequence of policy operators associated to the measurable minimizers of (B.1) for all $t \in \mathbb{N}$. For all $x \in \mathcal{X}$ it holds*

$$[(\mathsf{P}_{t+1} - \mathsf{P}_t)q(\mathsf{P}_t)](x) \geq 0 \tag{B.5}$$

*and*

$$J(\mathsf{P}_{t+1}) - J(\mathsf{P}_t) \geq 0 \tag{B.6}$$

*i.e. the objective function is always increased by a mirror descent iteration. Further, if $\tilde{q}(\mathsf{P}_t) \in B_b(\Omega)$ is such that $\|q(\mathsf{P}_t) - \tilde{q}(\mathsf{P}_t)\|_\infty \leq \varepsilon_t$, then (B.5) holds inexactly on $\tilde{q}(\mathsf{P}_t)$ as*

$$[(\mathsf{P}_{t+1} - \mathsf{P}_t)\tilde{q}(\mathsf{P}_t)](x) \geq -2\varepsilon_t. \tag{B.7}$$

*Proof.* By setting $\pi(x) = \pi_t(x)$ in (B.3), and recalling that $D(p; q) \geq 0$ with equality if and only if $p = q$, it follows that

$$\eta [(\mathsf{P}_{t+1} - \mathsf{P}_t)q(\mathsf{P}_t)](x) \geq D(\pi_t(x); \pi_{t+1}(x)) \geq 0,$$

giving (B.5). Integrating (B.5) over $(\mathsf{Id} - \gamma \mathsf{P}_{t+1}\mathsf{T})^{-*}\nu$ and using the Performance Difference Lemma A.3 one gets (B.6). Finally, we get (B.7) from

$$\begin{aligned}
[(\mathsf{P}_{t+1} - \mathsf{P}_t)\tilde{q}(\mathsf{P}_t)](x) &= [(\mathsf{P}_{t+1} - \mathsf{P}_t)q(\mathsf{P}_t)](x) + [(\mathsf{P}_{t+1} - \mathsf{P}_t)(\tilde{q}(\mathsf{P}_t) - q(\mathsf{P}_t))](x) \\
&\geq [(\mathsf{P}_{t+1} - \mathsf{P}_t)(\tilde{q}(\mathsf{P}_t) - q(\mathsf{P}_t))](x) \\
&\geq -\|(\mathsf{P}_{t+1} - \mathsf{P}_t)(\tilde{q}(\mathsf{P}_t) - q(\mathsf{P}_t))\|_\infty \\
&\geq -\|\mathsf{P}_{t+1} - \mathsf{P}_t\|\|\tilde{q}(\mathsf{P}_t) - q(\mathsf{P}_t)\|_\infty \\
&\geq -2\varepsilon_t.
\end{aligned} \tag{B.8}$$

Where the first inequality follows from (B.5), and the latter from the fact that policy operators are Markov operators and have norm 1, and $\|\mathsf{P}_{t+1} - \mathsf{P}_t\| \leq \|\mathsf{P}_{t+1}\| + \|\mathsf{P}_t\| = 2$. □

## B.1 Convergence rates of PMD

We are finally ready to prove the convergence rates for the Policy Mirror Descent algorithm (B.1). The proof technique is loosely based on [12, Theorem 8, Lemma 12], and extends them to the case of general state spaces through the key Lemma B.1 and using a fully operatorial formalism.

**Theorem 7** (Convergenge of Inexact PMD). *Let $(\pi_t)_{t \in \mathbb{N}}$ be a sequence of policies generated by Algorithm 1 that are all $(\mathcal{G}, \mathcal{F})$-compatible. If the action-value functions $\hat{q}_{\pi_t}$ are estimated with an error $\|q_{\pi_t} - \hat{q}_{\pi_t}\|_\infty \leq \varepsilon_t$, the iterates of Algorithm 1 converge to the optimal policy as*

$$J(\pi_*) - J(\pi_T) \leq \varepsilon_T + O\left(\frac{1}{T} + \frac{1}{T}\sum_{t=0}^{T-1} \varepsilon_t\right), \tag{16}$$

*where $\pi_* : \mathcal{X} \to \Delta(\mathcal{A})$ is a measurable maximizer of (8).*

*Proof.* As usual, in this proof we denote the estimated and exact action-value functions as $q_n(\mathsf{P}_t) := \hat{q}_{\pi_t} = (\mathsf{Id} - \gamma \mathsf{T}_n \mathsf{P}_t)^{-1} r_n$ and $q(\mathsf{P}_t) := q_{\pi_t} = (\mathsf{Id} - \gamma \mathsf{T}\mathsf{P}_t)^{-1} r$, respectively. From hypothesis, Algorithm 1 is well-defined since all policies it generates are $(\mathcal{G}, \mathcal{F})$-compatible. The resulting sequence of policies $(\pi_t)_{t \in \mathbb{N}}$ are generated via the update rule (14) on the inexact action-value functions $q_n(\mathsf{P}_t)$, as defined in (12). As the update rule (14) is a (measurable) minimizer of (B.1) when $D$ equals the Kullback-Leibler divergence, the three-points Lemma B.2 with $\pi(x) = \pi_*(x)$ yields

$$[(\mathsf{P}_* - \mathsf{P}_{t+1})q_n(\mathsf{P}_t)](x) \leq \frac{1}{\eta}D(\pi_*(x); \pi_t(x)) - \frac{1}{\eta}D(\pi_*(x); \pi_{t+1}(x)).$$

Adding and subtracting the term $[(\mathsf{P}_* - \mathsf{P}_{t+1})q(\mathsf{P}_t)](x)$, and bounding the remaining difference as $[(\mathsf{P}_* - \mathsf{P}_{t+1})(q(\mathsf{P}_t) - q_n(\mathsf{P}_t))](x) \le 2\varepsilon_t$ – see the derivation of (B.8) – one gets

$$[(\mathsf{P}_* - \mathsf{P}_{t+1})q(\mathsf{P}_t)](x) \le 2\varepsilon_t + \frac{1}{\eta}D(\pi_*(x); \pi_t(x)) - \frac{1}{\eta}D(\pi_*(x); \pi_{t+1}(x)).$$

Adding and subtracting $\mathsf{P}_t q(\mathsf{P}_t)$ on the left side gives

$$[(\mathsf{P}_* - \mathsf{P}_t)q(\mathsf{P}_t)](x) \le [(\mathsf{P}_{t+1} - \mathsf{P}_t)q(\mathsf{P}_t)](x) + 2\varepsilon_t + \frac{1}{\eta}D(\pi_*(x); \pi_t(x)) - \frac{1}{\eta}D(\pi_*(x); \pi_{t+1}(x)),$$

and integrating with respect to the positive measure $(\mathsf{Id} - \gamma\mathsf{P}_*\mathsf{T})^{-*}\nu$ and using the performance difference Lemma A.3 on the left hand side one has

$$
\begin{aligned}
J(\mathsf{P}_*) - J(\mathsf{P}_t) \le & \frac{1}{1-\gamma}\langle(\mathsf{P}_{t+1} - \mathsf{P}_t)q(\mathsf{P}_t) + 2\varepsilon_t, d_\nu(\mathsf{P}_*)\rangle \\
& \frac{1}{\eta(1-\gamma)}\langle D(\pi_*; \pi_t) - D(\pi_*; \pi_{t+1}), d_\nu(\mathsf{P}_*)\rangle,
\end{aligned}
\tag{B.9}
$$

where we used (A.8) on the right-hand-side terms. Since $(\mathsf{P}_{t+1} - \mathsf{P}_t)q(\mathsf{P}_t) + 2\varepsilon_t \ge 0$ because of (B.7), we can use fact (1) from Lemma A.5 with $(\mathsf{Id} - \gamma\mathsf{P}_{t+1}\mathsf{T})^{-1}$ and the performance difference Lemma A.3 to get

$$
\begin{aligned}
\langle(\mathsf{P}_{t+1} - \mathsf{P}_t)q(\mathsf{P}_t) + 2\varepsilon_t, d_\nu(\mathsf{P}_*)\rangle \le & \langle(\mathsf{Id} - \gamma\mathsf{P}_{t+1}\mathsf{T})^{-1}[(\mathsf{P}_{t+1} - \mathsf{P}_t)q(\mathsf{P}_t) + 2\varepsilon_t], d_\nu(\mathsf{P}_*)\rangle \\
= & \langle\mathsf{P}_{t+1}q(\mathsf{P}_{t+1}), d_\nu(\mathsf{P}_*)\rangle - \langle\mathsf{P}_t q(\mathsf{P}_t), d_\nu(\mathsf{P}_*)\rangle + \frac{2\varepsilon_t}{1-\gamma}.
\end{aligned}
$$

Substituting this bound in (B.9) and summing from $t = 0 \ldots T-1$ one gets to

$$
\begin{aligned}
\sum_{t=0}^{T-1} J(\mathsf{P}_*) - J(\mathsf{P}_t) \le & \frac{1}{1-\gamma}(\langle\mathsf{P}_T q(\mathsf{P}_T), d_\nu(\mathsf{P}_*)\rangle - \langle\mathsf{P}_0 q(\mathsf{P}_0), d_\nu(\mathsf{P}_*)\rangle) + \frac{2}{(1-\gamma)^2}\sum_{t=0}^{T-1}\varepsilon_t \\
& \frac{1}{\eta(1-\gamma)}\langle D(\pi_*; \pi_0) - D(\pi_*; \pi_T), d_\nu(\mathsf{P}_*)\rangle.
\end{aligned}
$$

Using facts (3) and (4) from Lemma A.5 we have that the terms $\langle\mathsf{P}q(\mathsf{P}), d_\nu(\mathsf{P}_*)\rangle$ on the right hand side can be bounded as

$$
\begin{aligned}
\langle\mathsf{P}q(\mathsf{P}), d_\nu(\mathsf{P}_*)\rangle = & \langle\mathsf{P}(\mathsf{Id} - \gamma\mathsf{T}\mathsf{P})^{-1}r, d_\nu(\mathsf{P}_*)\rangle \\
= & \langle(\mathsf{Id} - \gamma\mathsf{P}\mathsf{T})^{-1}\mathsf{P}r, d_\nu(\mathsf{P}_*)\rangle \\
= & \langle r, \mathsf{P}^*(\mathsf{Id} - \gamma\mathsf{P}\mathsf{T})^{-*}d_\nu(\mathsf{P}_*)\rangle \\
\text{(Duality)} \le & \|r\|_\infty \|\mathsf{P}^*(\mathsf{Id} - \gamma\mathsf{P}\mathsf{T})^{-*}d_\nu(\mathsf{P}_*)\|_{\text{TV}} \\
\text{(Lemma A.5 and } \|\nu\|_{\text{TV}} = 1) \le & \frac{\|r\|_\infty}{1-\gamma},
\end{aligned}
$$

while $-\langle D(\pi_*; \pi_T), d_\nu(\mathsf{P}_*)\rangle$ can be dropped due to the positivity of Bregman divergences yielding

$$\sum_{t=0}^{T-1} J(\mathsf{P}_*) - J(\mathsf{P}_t) \le \frac{2}{(1-\gamma)^2}\left(\|r\|_\infty + \sum_{t=0}^{T-1}\varepsilon_t\right) + \frac{1}{\eta(1-\gamma)}\langle D(\pi_*; \pi_0), d_\nu(\mathsf{P}_*)\rangle. \tag{B.10}$$

Now notice that for all $t < T$ it holds

$$
\begin{aligned}
J(\mathsf{P}_t) = & \langle\mathsf{P}_t q(\mathsf{P}_t), \nu\rangle \\
= & \langle\mathsf{P}_t q_n(\mathsf{P}_t), \nu\rangle + \langle\mathsf{P}_t(q(\mathsf{P}_t) - q_n(\mathsf{P}_t)), \nu\rangle \\
\text{(Equation (B.6))} \le & \langle\mathsf{P}_T q_n(\mathsf{P}_T), \nu\rangle + \langle\mathsf{P}_t(q(\mathsf{P}_t) - q_n(\mathsf{P}_t)), \nu\rangle \\
= & \langle\mathsf{P}_T q_n(\mathsf{P}_T), \nu\rangle + \langle\mathsf{P}_t(q(\mathsf{P}_t) - q_n(\mathsf{P}_t)), \nu\rangle + \langle\mathsf{P}_T(q_n(\mathsf{P}_T) - q(\mathsf{P}_T)), \nu\rangle \\
\le & J(\mathsf{P}_T) + \varepsilon_t + \varepsilon_T,
\end{aligned}
$$

so that

$$J(\mathsf{P}_*) - J(\mathsf{P}_T) \le \varepsilon_T + \frac{1}{T}\sum_{t=0}^{T-1} J(\mathsf{P}_*) - J(\mathsf{P}_t) + \varepsilon_t.$$

Combining this with (B.10) we obtain

$$J(\mathsf{P}_*) - J(\mathsf{P}_T) \le \varepsilon_T + \frac{1}{T}\left[\left(1 + \frac{2}{(1-\gamma)^2}\right)\sum_{t=0}^{T-1}\varepsilon_t + \frac{2\|r\|_\infty}{(1-\gamma)^2} + \frac{1}{\eta(1-\gamma)}\langle D(\pi_*; \pi_0), d_\nu(\mathsf{P}_*)\rangle\right],$$

leading to the desired bound. $\qquad\square$

## C  POWR Convergence Rates

In this section, we prove the convergence of POWR. To do so, we need to first show that under the choice of spaces $\mathcal{F}$ and $\mathcal{G}$ proposed in this work, the resulting PMD iterations are well defined. Then, we need to bound the approximation error of the estimates for the action-value functions of the iterates produced by the inexact PDM algorithm, which appear in the rates of Theorem 7.

### C.1  POWR  is Well-defined

In order to guarantee that the iterations of POWR  generate policies $\pi_t$ for which we can compute an estimator according to the formula in Proposition 3, we need to guarantee that all such policies are $(\mathcal{G}, \mathcal{F})$-compatible. In particular, we restrict to the case of the separable spaces introduced in Proposition 4, for which it turns out that it is sufficient to show that all policies belong to the space $\mathcal{H}$ characterizing $\mathcal{F} = \mathcal{H} \otimes \mathcal{H}$ and $\mathcal{G} = \mathbb{R}^{|\mathcal{A}|} \otimes \mathcal{H}$. The following results provide a candidate for choosing such a space.

**Theorem 5.** *Let $\mathcal{X} \subset \mathbb{R}^d$ be a compact set and let $\mathcal{H} = W^{2,s}(\mathcal{X})$ be the Sobolev space of smoothness $s > 0$ (see e.g. [38]). Let $\pi_t(a|\cdot)$ and $\hat{q}_{\pi_t}(\cdot, a)$ belong to $\mathcal{H}$ for any $a \in \mathcal{A}$ and $\pi_t(a|x) > 0$ for any $x \in \mathcal{X}$. Then the policy $\pi_{t+1}$ solution to the PMD update in (14) belongs to $\mathcal{H}$.*

*Proof.* We recall that Sobolev spaces [38] over a compact subset $\mathcal{X}$ of $\mathbb{R}^D$ are closed with respect to the operations of sum, multiplication, exponentiation or inversion (if the function is supported on the entire domain $\mathcal{X}$), namely for any two $f, f' \in \mathcal{H}$, $f + f'$, $ff'$, $e^f \in \mathcal{H}$ and, if $f(x) > 0$ for all $x \in \mathcal{X}$, $1/f \in \mathcal{H}$. This follows by applying the chain rule and the boundedness of derivatives over the compact $\mathcal{X}$ (see for instance [46, Lemma E.2.2]). The proof follows by observing that the one-step update $\pi_{t+1}$ in (14) is expressed precisely in terms of these operations and the hypothesis that $\pi_t(a|\cdot)$ and $\hat{q}_{\pi_t}(\cdot, a)$ belong to $\mathcal{H}$ for any $a \in \mathcal{A}$. □

Combining the choice of space $\mathcal{H}$ according to the above result and combining with the PMD iterations of Algorithm 1 we have the following corollary.

**Corollary 6.** *With the hypothesis of Proposition 4 let $\mathcal{H} = W^{2,s}(\mathcal{X})$ with $s > d/2$. Let $\mathsf{T}_n \in \mathsf{HS}(\mathcal{F}, \mathcal{G})$ and $r_n \in \mathcal{G}$ characterized as in Proposition 3. Let $\pi_0(a|\cdot) \propto e^{\eta q_0(\cdot, a)}$ for $q_0$ such that $q_0(\cdot, a) \in \mathcal{H}$ any $a \in \mathcal{A}$. Then, for any $t \in \mathbb{N}$ the PMD iterates $\pi_t$ generated by Algorithm 1 are such that $\pi_t(a|\cdot) \in \mathcal{H}$ and hence are $(\mathcal{G}, \mathcal{F})$-compatible.*

*Proof.* We proceed by induction. Since $\bar{q}(\cdot, a) \in \mathcal{H}$ we can apply the same reasoning in Theorem 5 to guarantee that $\pi_0(a|\cdot) \in \mathcal{H}$ for any $a \in \mathcal{A}$. Moreover, $\pi_0(a|\cot) > 0$ for any $a \in \mathcal{A}$ since it is the (normalized) exponential of a function. Hence $\pi_0$ is $(\mathcal{G}, \mathcal{F})$-compatible. Therefore, $\hat{q}_0$ obtained according to Proposition 3 is well defined and belongs to $\mathcal{G}$, implying $\hat{q}_0(\cdot, a) \in \mathcal{H}$ for any $a \in \mathcal{A}$. Now, assume by the inductive hypothesis that the policy $\pi_t(a|\cdot)$ generated by POWR at time $t$ and the corresponding estimator $\hat{q}_{\pi_t}(\cdot, a)$ of the action value function belong to $\mathcal{H}$ and that $\pi_t(a|x) > 0$ for any $(x, a) \in \Omega$. Then, by Theorem 5 we have that also $\pi_{t+1}$ the solution to the PMD update in (14) belongs to $\mathcal{H}$ (and is therefore $(\mathcal{G}, \mathcal{F})$-compatible). Additionally, since $\pi_{t+1}$ can be expressed as the softmax of a (finite) sum of functions in $\mathcal{H}$, we have also $\pi_{t+1}(a|x) > 0$ for al $(x, a) \in \Omega$, proving the inductive hypothesis and concluding the proof. □

The above corollary guarantees us that if we are able to learn our estimates for the action-value function in $\mathcal{H}$ a suitably regular Sobolev space, then POWR  is well-defined. This is a necessary condition to then being able to study its theoretical behavior in our main result.

### C.2  Controlling the Action-value Estimation Error

We now show how to control the estimation error for the action-value function. We start by considering the following application of the (generalized) Simulation lemma in Corollary A.7.

**Lemma 8** (Implications of the Simulation Lemma). *Let $\mathsf{T}_n$ and $r_n$ the empirical estimators of the transfer operator $\mathsf{T}$ and reward function $r$ as defined in Proposition 3, respectively. If $\mathsf{T}$*

*satisfies Assumption 1, $r \in \mathcal{G}$, and $\gamma \|\mathsf{T}_n\| < \gamma' < 1$, then, for every $(\mathcal{G}, \mathcal{F})$-compatible policy $\pi$*

$$\|\hat{q}_\pi - q_\pi\|_\infty \leq \frac{1}{1 - \gamma'} \left[ const_\psi \|r_n - r\|_\mathcal{G} + \frac{\gamma \|r\|_\infty}{1 - \gamma} \|\mathsf{T}|_\mathcal{F} - \mathsf{T}_n\|_{\mathsf{HS}} \right].$$

*Proof.* Recall that in the notation of these appendices, the action value of a policy and its estimator via the world model CME framework are denoted $q_\pi = q(\mathsf{P})$ and $\hat{q}_\pi = q_n(\mathsf{P})$ respectively. We can apply Corollary A.7 to obtain

$$q_n(\mathsf{P}) - q(\mathsf{P}) = (\mathsf{Id} - \gamma \mathsf{T}_n \mathsf{P})^{-1} (r_n - r) + \gamma (\mathsf{Id} - \gamma \mathsf{T}_n \mathsf{P})^{-1} (\mathsf{T} - \mathsf{T}_n) v(\mathsf{P}).$$

Then, by Lemma A.5, point 5, we have

$$\|q_n(\mathsf{P}) - q(\mathsf{P})\|_\infty \leq \frac{1}{1 - \gamma'} \left[ \|r_n - r\|_\infty + \gamma \|(\mathsf{T} - \mathsf{T}_n) v(\mathsf{P})\|_\infty \right],$$

where $v(\mathsf{P}) := \mathsf{P}(\mathsf{Id} - \gamma \mathsf{T} \mathsf{P})^{-1} r$ is the value function of the MDP, and we used that $\gamma \|\mathsf{T}_n\| < \gamma'$. Because of Assumption 1, $r \in \mathcal{G}$, and $\mathsf{P}$ being $(\mathcal{G}, \mathcal{F})$-compatible, it holds that $v(\mathsf{P}) \in \mathcal{F}$, while Proposition 3 implies $r_n \in \mathcal{G}$, and $(\mathsf{T} - \mathsf{T}_n) v(\mathsf{P}) \in \mathcal{G}$ as well. Therefore, using the reproducing property

$$\|r_n - r\|_\infty = \sup_{(x,a) \in \Omega} |\langle \psi(x,a), r_n - r \rangle_\mathcal{G}| \leq \|r_n - r\|_\mathcal{G} \sup_{(x,a) \in \Omega} \|\psi(x,a)\|_\mathcal{G} = C_\psi \|r_n - r\|_\mathcal{G}$$

where we assumed a bounded kernel $\langle \psi(x,a), \psi(x,a) \rangle \leq C_\psi$ for all $(x,a) \in \Omega$. Similarly, for the term depending on $\mathsf{T}_n - \mathsf{T}$ we have

$$\begin{aligned}
\|(\mathsf{T} - \mathsf{T}_n) v(\mathsf{P})\|_\infty &= \sup_{(x,a) \in \Omega} |[(\mathsf{T}|_\mathcal{F} - \mathsf{T}_n) v(\mathsf{P})](x,a)| \\
&= \sup_{(x,a) \in \Omega} |\langle \psi(x,a), (\mathsf{T}|_\mathcal{F} - \mathsf{T}_n) v(\mathsf{P}) \rangle_\mathcal{G}| \\
&= \sup_{(x,a) \in \Omega} \left| \mathrm{Tr} \left[ (v(\mathsf{P}) \otimes \psi(x,a)) (\mathsf{T}|_\mathcal{F} - \mathsf{T}_n) \right] \right| \\
&\leq \|\mathsf{T}|_\mathcal{F} - \mathsf{T}_n\|_{\mathsf{HS}} \sup_{(x,a) \in \Omega} |v(\mathsf{P})(x,a)| \\
&\leq \frac{\|r\|_\infty}{1 - \gamma} \|\mathsf{T}|_\mathcal{F} - \mathsf{T}_n\|_{\mathsf{HS}}.
\end{aligned}$$

Combining the previous two bounds, we get to

$$\|q_n(\mathsf{P}) - q(\mathsf{P})\|_\infty \leq \frac{1}{1 - \gamma'} \left[ C_\psi \|r_n - r\|_\mathcal{G} + \frac{\gamma \|r\|_\infty}{1 - \gamma} \|\mathsf{T}|_\mathcal{F} - \mathsf{T}_n\|_{\mathsf{HS}} \right],$$

as desired. $\qquad\square$

According to the result above, we can control the approximation error for the action value function in terms of the approximation errors $\|r_n - r\|_\mathcal{G}$ and $\|\mathsf{T}|_\mathcal{F} - \mathsf{T}_n\|_{\mathsf{HS}}$. This can be done by leveraging state-of-the-art statistical learning rates for the ridge regression and CME estimators from [40, 21, 47]. The following lemma connects Assumption 2 with the notation used in [40] which enables us to use the required result.

**Lemma C.1** (Relation between (A.8) and [40]'s definition). *The following two facts are equivalent*

1. *$g \in \mathcal{G}$ satisfies the strong source condition Assumption 2 with parameter $\beta$ on the probability distribution $\rho$.*

2. *$g \in [\mathcal{G}]_\rho^{1+2\beta}$ as in the notation of [40].*

*Proof.* Using the same notations as in [40], we have

$$g \in [\mathcal{G}]_\rho^{1+2\beta} \iff g = \sum_{i \in \mathbb{N}} a_i \mu_i^{\frac{1}{2} + \beta} e_i \text{ and } \|(a_i)_{i \in \mathbb{N}}\|_{\ell^2} < \infty.$$

For $1 \implies 2$ we have that that for $g \in [\mathcal{G}]_\rho^{1+2\beta}$

$$C_\rho^{-\beta} g = \sum_{i \in \mathbb{N}} a_i \mu_i^{\frac{1}{2}} e_i \tag{C.1}$$

whose $\mathcal{G}$-norm is $\left\| C_\rho^{-\beta} g \right\|_{\mathcal{G}} = \left\| (a_i)_{i \in \mathbb{N}} \right\|_{\ell^2} < \infty$.

For $2 \implies 1$, let $g = \sum_{i \in \mathbb{N}} b_i \mu_i^{\frac{1}{2}} e_i$ with $\left\| (b_i)_{i \in \mathbb{N}} \right\|_{\ell^2} < \infty$ (since). Now, $\left\| C_\rho^{-\beta} g \right\|_{\mathcal{G}} < \infty$ is equivalent to $\left\| (b_i \mu_i^{-\beta})_{i \in \mathbb{N}} \right\|_{\ell^2} < \infty$. By letting $b_i = \mu_i^\beta a_i$ we have that $\left\| (a_i)_{i \in \mathbb{N}} \right\|_{\ell^2} < \infty$ and that

$$g = \sum_{i \in \mathbb{N}} a_i \mu_i^{\frac{1}{2}+\beta} e_i,$$

that is $g \in [\mathcal{G}]_\rho^{1+2\beta}$. $\qquad\square$

With the connection between [40] and Assumption 2 in place we can characterize the bound on the approximation error for the world model-based estimation of the action-value function.

**Proposition C.2.** *Let* $\mathsf{T}_n$ *and* $r_n$ *the empirical estimators of the transfer operator* $\mathsf{T}$ *and reward function* $r$ *as defined in Proposition 3, respectively. When* $\mathsf{P}$ *is a* $(\mathcal{G}, \mathcal{F})$-*compatible policy as in Definition 1 and the strong source condition Assumption 2 is attained with parameter* $\beta$, *it holds*

$$\left\| q_n(\mathsf{P}) - q(\mathsf{P}) \right\|_\infty \le O(\delta^2 n^{-\alpha}), \tag{C.2}$$

*with rates* $\alpha \in \left( \frac{\beta}{2+2\beta}, \frac{\beta}{1+2\beta} \right)$ *and probability not less than* $1 - 4e^{-\delta}$.

*Proof.* We use Lemma C.1 to apply Theorem 3.1 (ii) from [40] to show that under Assumption 2 with parameter $\beta$ it holds, with probability not less than $1 - 4e^{-\delta}$,

$$\left\| r_n - r \right\|_{\mathcal{G}} \le \delta^2 c_r n^{-\alpha_r}. \tag{C.3}$$

The rate $\alpha_r \in \left( \frac{\beta}{2+2\beta}, \frac{\beta}{1+2\beta} \right)$ is determined by the properties of the inclusion $\mathcal{G} \hookrightarrow B_b(\Omega)$, and the constant $c_r > 0$ is independent of $n$ and $\delta$. Similarly, point (2.) of [21, Theorem 2] shows that under Assumption 2

$$\left\| \mathsf{T}|_{\mathcal{F}} - \mathsf{T}_n \right\|_{\mathsf{HS}(\mathcal{F}, \mathcal{G})} \le \delta^2 c_\mathsf{T} n^{-\alpha_\mathsf{T}} \tag{C.4}$$

again with probability not less than $1 - 4e^{-\delta}$, rates $\alpha_\mathsf{T} \in \left( \frac{\beta}{2+2\beta}, \frac{\beta}{1+2\beta} \right)$ and with $c_\mathsf{T} > 0$ independent of $n$ and $\delta$. Combining every bound and denoting $\alpha := \min(\alpha_r, \alpha_\mathsf{T})$, we conclude

$$\left\| q_n(\mathsf{P}) - q(\mathsf{P}) \right\|_\infty \le \frac{\delta^2}{1 - \gamma'} \left[ C_\psi c_r + \frac{\gamma c_\mathsf{T} \|r\|_\infty}{1 - \gamma} \right] n^{-\alpha} = O(\delta^2 n^{-\alpha}). \tag{C.5}$$

as required. $\qquad\square$

### C.3 Convergence Rates for POWR

With a bound on the estimation error of the action-value function by Algorithm 1, we are finally ready to state the complexity bounds for POWR.

**Theorem 9.** *Let* $(\pi_t)_{t \in \mathbb{N}}$ *be a sequence of policies generated by Algorithm 1 in the same setting of Corollary 6. If the action-value functions* $\hat{q}_{\pi_t}$ *are estimated from a dataset* $(x_i, a_i; x_i')_{i=1}^n$ *with* $(x_i, a_i) \sim \rho \in \mathcal{P}(\Omega)$ *such that Assumption 2 holds with parameter* $\beta$, *the iterates of Algorithm 1 converge to the optimal policy as*

$$J(\pi_*) - J(\pi_T) \le O\left( \frac{1}{T} + \delta^2 n^{-\alpha} \right)$$

*with probability not less than* $1 - 4e^{-\delta}$. *Here,* $\alpha \in \left( \frac{\beta}{2+2\beta}, \frac{\beta}{1+2\beta} \right)$ *and* $\pi_* : \mathcal{X} \to \Delta(\mathcal{A})$ *is a measurable maximizer of* (8).

*Proof.* Since the setting of Corollary 6 implies that $\mathsf{P}_t$ are $(\mathcal{G}, \mathcal{F})$-compatible for all $t$, and Assumption 2 is holding, then $q(\mathsf{P}_t)$ and $q_n(\mathsf{P}_t)$ belong to $\mathcal{G}$ for all $(\mathsf{P}_t)_{t \in \mathbb{N}}$. This assures that we can use the statistical learning bounds Proposition C.2 into Theorem 7, yielding the final bound. $\qquad\square$

# D Experimental details

## D.1 Additional Results

In Fig. 2 we show the average timestep at which a reward threshold is met during the training phase. The testing environments are the same as introduced previously, with reward thresholds being the standard ones given in [26], except for the Taxi-v3 environment, where it is marginally lower. Interestingly, in this environment, only DQN and our algorithm are capable of achieving the original threshold within $1.5 \times 10^6$ timesteps during the training. On the other hand, the new lower threshold is also reached by the PPO algorithm.

Our approach attain the desired reward quicker than the competing algorithms. Furthermore, the timestep at which POWRreaches the threshold exhibits a lower variance compared to other techniques. This implies that our approach requires a stable amount of timesteps to learn how to solve a specific environment.

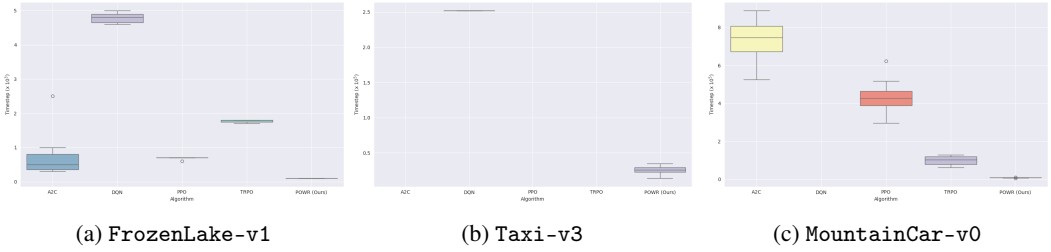

(a) FrozenLake-v1    (b) Taxi-v3    (c) MountainCar-v0

Figure 2: Mean timestep at which various algorithms attain a specified reward threshold during their training. The reward targets are set at $0.8$ for FrozenLake-v1, $6$ for Taxi-v3, and $-110$ for MountainCar-v0. The absence of a box indicates that the corresponding algorithm was unable to meet the reward threshold within the training process.

## D.2 Other methods

We compare the performance of our algorithm with several baselines. In particular, we considered A2C [41], DQN [4], TRPO [7] and PPO [6], which we implemented using the stable baselines library [48]. We used the standard hyperparameters in [48].

