# OpenReview forum: "Operator World Models for Reinforcement Learning"
_NeurIPS.cc/2024/Conference — NeurIPS 2024 poster_

### Official Review · Reviewer_Tuc7 · 2024-07-03

**Soundness:** 4
**Presentation:** 4
**Contribution:** 4
**Rating:** 8
**Confidence:** 4

**Summary:**

This paper extends policy mirror descent to the RL setting. They solve the problem of requiring a exact value function by formulating an approximate value function based on operators over the transition and reward functions, which yields a closed form solution. Then they use this approximate value function for policy improvement. Lastly, they provide theoretical analysis on convergence guarantees and error bounds while explicitly laying out assumptions. They also perform experiments on toy RL problems and show that their algorithm outperforms typical RL baselines on these low-dimensional problems.

**Strengths:**

1) This paper's presentation is great. How this work relates to prior work is clearly discussed. Key concepts from this paper are well explained and well motivated.
2) Theoretical analysis is sound and shows promising results.
3) Comparison to RL baselines, even on toy problems, is much appreciated. Results are encouraging.
4) Discussions of limitations and assumptions are detailed.
5) Key results of this paper, IE value functions with closed form solutions via operators, are novel and significant.

**Weaknesses:**

1) I would like to see experiments on more complicated MDPs, with large state-action spaces, to test the scaling properties. I recognize that this paper already makes large contributions, so this is not necessary for now, but it is a great future direction.
2) I would like to see extensions to continuous action spaces. Again not necessary for now, but it seems like it should be possible given the formulation with additional approximations.

Minor:
Typo on line 335

**Questions:**

How well can this algorithm scale to higher-dimensional settings?

**Limitations:**

Authors have adequately addressed limitation and assumptions.

I see no potential negative societal impacts as a result of this work.

---

> ### Author Rebuttal · Authors · 2024-08-06
>
> We thank the reviewer for the positive feedback.
>
> - __Experiments on more complicated MDPs__: we agree with the reviewer and we are very excited to test our methods on more complicated environments. As the reviewer pointed out, our main focus in this work was to prove the novel theoretical contributions of Prop. 3, Cor. 7, and Thm. 9. Our experiments in section 4 are meant as empirical support to our theoretical findings. As we discussed in the conclusions of this work, competitive evaluation of POWR on richer MDPs requires additional research on:
> Scaling the method to high-dimensional settings (see also our additional comments below)
> Combining POWR with deep representation learning schemes to learn the operator-world model beyond settings suited for Sobolev representations (e.g. images).
>
> - __POWR in Continuous Action Spaces__: that is a very interesting question. We refer to our global reply here on OpenReview since this was a topic of interest also for other reviewers.
>
> - __Scaling to high dimensional settings__: breaking down this question into the computational and statistical perspectives:
>     - _Statistics_: from the statistical perspective (namely Thm. 9), the method is not (directly) affected by the dimensionality of the problem. The learning rates in Thm. 9 mainly depend on the norms of the operator world model and reward function as elements of the feature space where learning is carried out (cf. the final bound reported in Thm. A.11 in the appendix). These norms can be interpreted as capturing “how well suited” the chosen feature space is to represent the operator world model.
>
>         To (partially) answer the reviewer’s question, in the case of the Sobolev feature spaces used in this work, there exists a relation between the dimensionality of the ambient space (namely the state space) and the norm of a function in terms of its smoothness (see [28,29]). In other words - and very loosely speaking - the larger the space dimension, the smoother the target function needs to be to maintain a small norm. This means that if the reward function or the transition operator are not very smooth, we might incur large constants in Thm. 9, hence slower rates. Choosing the hypothesis space is key to fast convergence, and we have discussed how to extend POWR with other hypothesis spaces than Sobolev spaces in our conclusion and future work.
>
>         This behavior is well-understood, albeit rather technical, in the kernel learning literature for traditional Sobolev spaces (for instance, see Sec. 3 and Ex. 3 in [B] or combining Chapter 5 in [28] with the results on interpolation spaces from [C] or [D]. References provided below). It is less clear how this interpretation will extend to less traditional feature spaces. We thank the reviewer for the question. We will add the discussion above as a remark following Thm. 9.
>
>     - _Computations_: by leveraging the kernel trick argument from Prop. 3, the dimensionality of the ambient space or the feature space does not come into play. In contrast, computations are affected by the number of samples observed (as is the case with most kernel methods). However, as observed in the recent kernel literature, this issue can be easily mitigated by adopting random projection approaches such as Nystrom sampling [25,26], significantly reducing computations without compromising model performance.
>
> - __Typos__: Thanks for pointing out typos.
>
> __Additional references__
>
> [B] Cucker and Smale. "On the mathematical foundations of learning." Bulletin of the American mathematical society, 2002.
>
> [C] Steve and Zhou. "Learning theory estimates via integral operators and their approximations." Constructive approximation, 2007.
>
> [D] Caponnetto and De Vito. "Optimal rates for the regularized least-squares algorithm." Foundations of Computational Mathematics, 2007.

---

> > ### Comment · Reviewer_Tuc7 · 2024-08-07
> >
> > Thank you for your comments.

---

### Official Review · Reviewer_41Mf · 2024-07-13

**Soundness:** 2
**Presentation:** 3
**Contribution:** 3
**Rating:** 7
**Confidence:** 3

**Summary:**

This paper proposes an approach (the first PMD approach adapted to RL setting) that can learn a world model using conditional mean embeddings (CME). The operatorial formulation of RL is used to express the action-value function in closed form via matrix operations. The proposed algorithm is proved to converge to global optimum, and validated by simple/preliminary experiments.

**Strengths:**

To the best of my knowledge, this work is the first one to apply PMD to RL setting with provable global convergence guarantee.

The references and previous work are cited and well-discussed.

The theoretical contribution of this work is sound and solid.

The proposed method is validated to be more sample efficient than the current state-of-the-art RL algorithms like PPO, DQN, TRPO

**Weaknesses:**

Line 88: "Borrowing from the convex optimization literature", please give the exact reference.

Line 97: at the end, lack of a "space"

From Line 91 to the end of the manuscript..., it seems like the hyper-link of references does not work. They lead me to incorrect place, and make it hard to follow the idea.

The experimental validation is limited to simple/toy environments in OpenAI Gym.

In the caption of Figure 1, I would change "dark lines" to "solid line". The authors should also mention the reward threshold is the "success" reward in the caption, not just in the main texts. It is probably better to add the lable of dashed line in the plot.

Since the Figure 1 is in logscale, the authors should show the learning curves starting from 0 or 10^1, so that we can observe the full learning process.

**Questions:**

In Figure 1, it is a bit surprising that the variance of the proposed method in (a) and (b) is 0 (or almost 0). I know that general RL algorithms will still have some fluctuation. Maybe the authors can give some intuition why the proposed method achieves such small variance?

In Figure 1(c), why the proposed method stops at 10^4 step? I would expect to see the full learning curve, otherwise it is not convincing that the algorithm is convergent.

Does POWR work in continuous action space? Even simple tasks in Gym including Pendulum, MountainCarContinuous, LunarLanderContinuous, etc.

**Limitations:**

The major limitation of this work is the empirical validation, where the authors only consider three toy/simple environments in OpenAI Gym.

Also, the experiments are limited to the discrete action space. It will be better to see if the proposed algorithm works in continuous action space like robotics, etc.

---

> ### Author Rebuttal · Authors · 2024-08-06
>
> We thank the reviewer for the feedback and kind words.
>
> - __Line 88 - references__: the requested references are provided in the same sentence referred to by the reviewer, line 90. They are references [19,20] in the paper, namely:
>
>     [19] Beck and Teboulle. Mirror descent and nonlinear projected subgradient methods for convex optimization. Operations Research Letters, 2003
>
>     [20] Sébastien Bubeck, Convex optimization: Algorithms and complexity. Foundations and Trends in Machine Learning, 2015
>
> - __X-axis Range for Plots__: In Fig. 1, we reported the average reward starting from 10^3 because, for most environments, the region $[0,  10^3]$ did not provide much information. However, we agree with the reviewer that the full plot is useful to have a complete picture. Please refer to the pdf attached to our global post here on OpenReview.
>
> - __Variance in Figure 1 (a) and (b)__: Both FrozenLake and Taxi are simple deterministic tabular environments. Once POWR has collected enough evidence about the reward function and the transition operator, there are no fluctuations anymore: the mean reward is constant, and the variance converges to zero. This is precisely the advantage of learning an operator-based world model rather than adopting an implicit one from which we can only sample possible trajectories. We also point out that the behavior is different for MountainCar, since the environment has infinitely many states, and POWR still incurs errors in the approximation of the transition operator.
>
> - __Fig. 1 (c) stops at ~10k steps__: Thanks for pointing this out. Indeed, in the original figure the convergence of POWR for MountainCar was not fully evident from the plot. We have re-run the experiment for a larger number of epochs. See the pdf attached to our global post here on OpenReview. Given the short time for the rebuttal period, we run POWR collecting over 40K samples. The new results show that POWR retains its performance after reaching the “success” threshold, suggesting that it achieved empirical convergence.
>
> - __POWR in Continuous Action Spaces__: that is a very interesting question. We refer to our global reply here on OpenReview since this was a topic of interest also for other reviewers.
>
> - __Other feedback__: We thank the reviewer for pointing out typos and other issues/suggestions. We will amend the paper accordingly. (Concerning the broken hyperlinks, it was due the way we separated the paper from the appendices/supplementary material. Hyperlinks, indeed, work as intended in the full paper pdf provided in the supplementary material uploaded in the original submission).

---

> > ### Comment · Reviewer_41Mf · 2024-08-08
> > **Rebuttal**
> >
> > Thank you for providing the updated Figure 1. It indeed looks more reasonable.
> >
> > But is there a reason not to run 10^6 timesteps for MountainCar? I understand that rebuttal has limited time, but MountainCar is a very easy problem, which should not take that long. Will the authors plot 10^6 timesteps in the final version of the paper?

---

> > > ### Author Response · Authors · 2024-08-08
> > >
> > > We thank the reviewer for the prompt reply.
> > >
> > > Yes, we will align with the other baselines in the final version of the paper, running experiments up to 10^6 steps, as stated in the attached PDF.
> > >
> > > Given the theoretical nature of this work and the limited time available, we have not thoroughly optimized our implementation of POWR, which resulted in longer runtimes. While there is significant room for optimization—and we are excited to advance POWR further— we would like to highlight that the main focus of this work was to prove the novel theoretical contributions of Proposition 3, Corollary 7, and Theorem 9. Our experiments demonstrate POWR's statistical efficiency, achieving success at least an order of magnitude faster than competitors.

---

> > > > ### Comment · Reviewer_41Mf · 2024-08-12
> > > > **Rebuttal2**
> > > >
> > > > Thanks for your clarification.
> > > > Based on the discussions so far, most of my concerns have been (or will be) addressed. At this stage, I will increase the score. Thanks!

---

### Official Review · Reviewer_68rN · 2024-07-13

**Soundness:** 3
**Presentation:** 3
**Contribution:** 3
**Rating:** 5
**Confidence:** 2

**Summary:**

(This review has been updated after revert of desk reject)

**Strengths:**

(updated after revert of desk reject)
It's a solid theoretical paper with nice writing and experimental results on continuous control benchmarks.

**Weaknesses:**

(updated after revert of desk reject)
The experimental results and more interpretation/justification of the results are limited. It would be stronger if the authors could further connect the experiments with the theoretical results.

**Questions:**

N/A

**Limitations:**

yes

---

> ### Author Rebuttal · Authors · 2024-08-06
>
> We kindly point out to the reviewer that the decision on desk rejection was reverted by the Program Chairs
>
> (The initial desk rejection had been due to the automated checker failure to detect the NeurIPS checklist in the supplementary material. This happened to several papers this year).
>
> Should the reviewer have any questions, we’d be glad to answer them.

---

> > ### Comment · Reviewer_68rN · 2024-08-14
> >
> > Thanks authors for the reminder. I updated the review and it's short, but I'm holding (weakly) a positive opinion.

---

> > > ### Author Response · Authors · 2024-08-14
> > >
> > > We thank the reviewer for their feedback.
> > >
> > > We appreciate the reviewer’s suggestion to enhance the connection between our experimental results and the theoretical analysis. In the experimental section (lines 367/368), we have already mentioned that our method, POWR, converges to a global optimum (as demonstrated in Theorems 7 and 9), and exhibits smaller sample complexity compared to other baselines, which aligns with the expected behavior of world models. According to the reviewer’s suggestion, we will make these connections clearer.
> > >
> > > Additionally, we would like to emphasize that the main contributions of our paper are theoretical. In particular, many highly influential works in this area such as [10, 11, 12] do not include an experimental section at all. Nonetheless, we included experiments to provide additional support for our theoretical claims. While the experimental section serves as a supplementary validation, our primary focus remains on the theoretical advancements presented in the paper.

---

### Official Review · Reviewer_vreF · 2024-07-19

**Soundness:** 3
**Presentation:** 3
**Contribution:** 3
**Rating:** 7
**Confidence:** 3

**Summary:**

The paper presents a practical implementation of Policy Mirror Descent (PMD) for Reinforcement Learning (RL). PMD requires knowledge of the action value function of the current policy at each iteration. Existing methods that approximate the action-value function depend on the ability to restart the Markov Decision Process (MDP) from each state multiple times. Instead, this paper proposes learning estimators of the reward function and the transition model and combining them to approximate the action-value function. To facilitate this, an RL formulation using operators is introduced, enabling the use of conditional mean embeddings for approximation. The authors study the convergence of the proposed algorithm to the global optimum and compare it with other RL methods on classic Gym environments.

**Strengths:**

- The paper is well-written, and the authors thoroughly introduce their method in relation to existing techniques.

**Weaknesses:**

- Considering the connection between the proposed method and world models, the authors should discuss world models more thoroughly [1-3].

[1] Schmidhuber, J. Making the world differentiable: On using supervised learning recurrent neural
networks for dynamic reinforcement learning and planning in non-stationary environments. (1990)

[2] Schmidhuber, J. An on-line algorithm for dynamic reinforcement learning and planning in reactive environments. (1990)

[3] Ha, D., and Schmidhuber, J. World Models (2018)

**Questions:**

See above.

The paper is rather technical, and several sections are beyond my expertise. I am looking forward to discussing it further with other reviewers to potentially increase the score for this submission.

**Limitations:**

Limitations are discussed in the paper.

---

> ### Author Rebuttal · Authors · 2024-08-06
>
> We thank the reviewer for the positive review.
>
> In particular, we thank the reviewer for the additional references. We will add them to the paper with the following expanded discussion in the introductory section of our paper, on line 42 (we denoted with R1,R2,R3 the references suggested by the reviewer):
>
> _The notion of world models for RL has been introduced by Ha and Schmidhuber in [R3] (building on ideas from [R1,R2]), where RNNs are used to learn the transition probability of the MDP. Traditional world model methods such as those proposed in [R3,16] emphasize learning an implicit model of the environment in the form of a simulator. The simulator can be sampled directly in the latent representation space, which is usually of moderate dimension, resulting in a compressed and high-throughput model of the environment._

---

> > ### Comment · Reviewer_vreF · 2024-08-08
> > **Official response**
> >
> > Thank you for your response. Please note that the claim "The notion of world models for RL was introduced by Ha and Schmidhuber in [R3]" is incorrect. World models in RL were already introduced in Reference R1 (1990) and in Dyna ([4], 1991).
> >
> > I trust that the authors will accurately discuss the background of world models. Having read the other reviews and the authors' responses, I am happy to increase the score for this submission.
> >
> > [4] Sutton, R., Dyna, an integrated architecture for learning, planning, and reacting. (1991)

---

> > > ### Author Response · Authors · 2024-08-08
> > >
> > > We thank the reviewer for the feedback and for raising their original score. We will update the introduction to accurately reflect the prior work on world models in RL, as the reviewer suggested.

---

### Author Rebuttal · Authors · 2024-08-06

Thanking all reviewers for their feedback on our work, in this global reply we share our insights on the extension of our results to continuous action spaces, a question shared among reviewers. In addition, in the attached pdf we report Figure 1 updated according to the suggestions of reviewer __41Mf__.

Extending POWR to action spaces of infinite cardinality could in principle be naively tackled by approximating the normalization integral in the denominator of eq. (13) via sampling methods such as Monte Carlo (MC) sampling. This approach, however, poses two questions:

- _Computational_: the integral approximation is conditional to the input state, implying that whenever we need to evaluate the policy on a new state, a new approximation is required. Unrolling this computation backward through all PMD iterations is potentially computationally expensive.

- _Approximation_: it is not clear whether this is a good approximation of the ideal solution to the PMD step in equation (2) and how it will impact the convergence rates proved in the theoretical section.

This makes the naive approach to extending POWR via MC sampling potentially limited.

An alternative promising approach is to approximate the solution to the PMD step in equation (2) via an iterative optimization strategy. This would still introduce approximation errors, but they could be controlled via the optimization rates for mirror descent. In particular, the recent  work in [A]

[A] _Aubin-Frankowski, Korba, and Léger. "Mirror descent with relative smoothness in measure spaces, with application to sinkhorn and em." Advances in Neural Information Processing Systems 35 (2022): 17263-17275._

shows that it is possible to cast and perform mirror descent also on general probability spaces (a question that was still open) maintaining analogous convergence rates to the finite setting. To our preliminary investigation, adapting the analysis in [A] to Policy Mirror Descent is feasible. However, the open questions in this sense are:
How to concretely perform the optimization of eq. (2) in practice, and
Whether this additional iterative procedure would compromise the computational efficiency of POWR’s pipeline.

We hope that these comments provide some additional context to the question posed by the reviewers, and why continuous action spaces ended up not being addressed in the paper at this time.

---

### Decision · Program_Chairs · 2024-09-25

**Decision:**

Accept (poster)

**Comment:**

The paper presents a practical implementation of Policy Mirror Descent (PMD) for Reinforcement Learning (RL), which ordinarily requires knowledge of the action value function of the current policy at each iteration. This paper proposes learning estimators of the reward function and the transition model and combining them to approximate the action-value function. An RL formulation using operators is introduced, enabling the use of conditional mean embeddings for approximation. The authors study the convergence of the proposed algorithm and compare it with other RL methods on classic Gym environments.
Overall all reviewers are quite enthusiastic about the paper and I am glad to recommend acceptance.